# DEPs Induce Local Ige Class Switching Independent of Their Ability to Stimulate iBALT de Novo Formation

**DOI:** 10.3390/ijerph192013063

**Published:** 2022-10-11

**Authors:** Dmitrii Borisovich Chudakov, Mariya Vladimirovna Konovalova, Elena Igorevna Kashirina, Olga Dmitrievna Kotsareva, Marina Alexandrovna Shevchenko, Daria Sergeevna Tsaregorodtseva, Gulnar Vaisovna Fattakhova

**Affiliations:** 1Laboratory of Cell Interactions, Shemyakin-Ovchinnikov Institute of Bioorganic Chemistry, 16/10 Miklukho-Maklaya Str., Moscow 117997, Russia; 2Faculty of Medical Biology, Sechenov First Moscow State Medical University, 2 Bolshaya Pirogovskaya Str., Moscow 1194535, Russia

**Keywords:** diesel particulate matter, antibody production, tertiary lymphoid structures, local Ig class switch, antigen doses, lungs

## Abstract

Background: Diesel exhaust particles (DEPs) are leading to a general increase in atopic diseases worldwide. However, it is still unknown whether DEPs induce systemic B-cell IgE class switching in secondary lymphoid organs or locally in the lungs in inducible bronchus-associated lymphoid tissue (iBALT). The aim of this work was to identify the exact site of DEP-mediated B-cell IgE class switching and pro-allergic antibody production. Methods: We immunized BALB/c mice with different OVA doses (0.3 and 30 µg) intranasally in the presence and absence of two types of DEPs, SRM1650B and SRM2786. We used low (30 µg) and high (150 µg) DEP doses. Results: Only a high DEP dose induced IgE production, regardless of the particle type. Local IgE class switching was stimulated upon treatment with both types of particles with both low and high OVA doses. Despite the similar ability of the two standard DEPs to stimulate IgE production, their ability to induce iBALT formation and growth was markedly different upon co-administration with low OVA doses. Conclusions: DEP-induced local IgE class switching takes place in preexisting iBALTs independent of de novo iBALT formation, at least in the case of SRM1650B co-administered with low OVA doses.

## 1. Introduction

One of the main phenomena hypothesized as explaining the increase in the prevalence of different IgE-mediated pathologies such as asthma, allergic rhinitis, and atopic dermatitis is air pollution [1]. The main type of pollutant that is linked to allergy and asthma development is particulate matter, especially diesel exhaust particles (DEPs) [2,3]. These are by-products of diesel fuel combustion and consist of a carbonaceous core with adsorbed polycyclic aromatic hydrocarbons (PAHs), as well as other organic contaminants (for example aliphatic hydrocarbons, diphenyl esters), and nonorganic components (aluminum, heavy metals, and others) [4]. These organic compounds possess proinflammatory properties [3,5,6]. Despite the tightening of fuel regulations, aimed at minimizing the formation of such particles during combustion, the concentrations of these particles in the air of large metropolitan areas are often high enough to stimulate biological responses [7,8]. According to some previous studies, the probability of asthma development in human subjects is closely related to the intensity of DEP exposure, especially in children [9,10,11]. However, the associations between the intensity of DEP exposure and the development of asthma in adults are not strong [9], and a recent study indicated that DEPs mostly induce immune-cell activation in people with established asthma rather than in healthy subjects [12]. It was shown that DEP exposure in early childhood may lead to asthma development in adulthood [13]. DEPs also seem to be related to the development of certain cardiovascular [14] and neurodegenerative diseases [15]. As shown by several research groups, the DEP concentration in the ambient air is strongly associated with an increasing prevalence of asthma cases [2,3,16,17]. The administration of these particles in a mouse allergy model leads to the exacerbation of allergic inflammation and pro-allergic antibody production [5,18,19,20,21,22,23]. The main mechanisms responsible for DEPs’ activity are linked to the induction of oxidative stress and cell damage [9,24], which induce the release of proinflammatory mediators from the bronchial epithelium and stimulate cytokine production by T-helper cells. It is likely that moderate DEP levels induce Th-2-related inflammation, while high DEPs levels induce Th-17-related inflammation [9].

Recently, it was shown that the intratracheal instillation of monodisperse alum or silica particles caused iBALT formation and local IgE production in C57BL/6 mice [25]. Although DEP-induced allergic inflammation has been studied in detail [5,18,19,20,21,22,23], the exact location of DEP-induced B-cell class switching and IgE production has not yet been identified. Although the data confirmed that DEPs induced local lymphoid tissue formation in mouse lungs upon instillation [26], it remains unclear whether IgE production could be functionally associated with these structures. In contrast with monodisperse alum or silica, DEPs are multicomponent and polydisperse in nature [4]. The toxic effects of DEPs on living cells are mostly linked to the oxidative and genotoxic stress induced by PAHs, especially by adsorbed benzo(a)pyrene (BaP), although the induction of cell death cannot be excluded [3,22,27,28,29,30]. It is noteworthy that despite the inability of bulk DEPs to penetrate the lung epithelium and enter the systematic circulation, fine DEPs less than 0.1 µm in diameter can do this [2,3]. Upon DEPs’ accumulation in lung tissue, the PAHs are able to desorb from the carbon core and enter the blood or lymphatic system. Therefore, DEPs accumulated in the lung tissue may have not only local but also systemic effects on immunity.

The physical properties of DEPs vary significantly depending on their specific sources as well as the atmospheric conditions and the time they spend in the air [2,22]. For example, SRM1650B (DEP1) collected directly from the heat exchangers of a diesel engine have a 0.18 µm mean diameter when disaggregated by ultrasound, but when they were aggregated, their resulting size exceeded 1–10 µm [31]. SRM2786 (DEP2) collected from the air of urban areas in Central Europe has an average particle diameter of 2.8 µm without preliminary disaggregation (and due to this fact, they are considered fine particulate matter), with 10% of the particles being smaller than 0.91 µm and 90% being smaller than 6.9 µm [32]. However, the ability to enter the lungs after inhalation is inversely proportional to the size. It is widely accepted that particles smaller than 2.5 µm have a high ability to penetrate lung tissue after intranasal inhalation. Particles smaller than 0.1 µm can cross epithelial barriers [2,3]. A DEP2 sample contained hardly any such particles, despite its particles being classed as “fine”. This is in contrast to the case of a DEP1 sample, in which 10% of the particles were smaller than 0.12 µm, according to a NIST analysis [31,32]. The chemical composition of two type of DEPs also vary to some extent, but these differences are not so crucial as their differences in size and ability to form aggregates [31,32].

In our previous work, we found that low (but not high) antigen doses induced rapid local B-cell IgE class switching, accompanied by minimal IgG1 production in fat-associated lymphoid clusters rather than regional lymph nodes [33]. However, the properties of iBALT may differ significantly from those of fat-associated lymphoid structures. Therefore, it is not clear whether low or high antigen doses co-administered with DEPs would stimulate IgE class switching.

The aim of this work was to clarify (a) whether DEPs induced local (in lung tissue) or systemic (in lymph nodes) IgE class switching and IgE production when introduced with either low or high antigen doses and (b) whether local IgE class switching was accompanied by iBALT induction. We also compared the effects of two types of DEPs, SRM1650B (DEP1) and SRM2786 (DEP2), on the local and systemic immune responses.

## 2. Materials and Methods

### 2.1. Animals

Female BALB/c mice, 5–6 weeks old, were purchased from the Scientific Center of Biomedical Technologies (Andreevka, Russia). The mice were housed in SPF conditions for two weeks before the experiments. During all the experimental procedures, the mice were maintained under a 12 h light/dark cycle at room temperature (20–23 °C) at 40–60% humidity in a special room in a vivarium. The mice were housed in plastic cages (10–12 mice per cage) with wood shavings as a bedding material. They were fed ad libitum with granular feed. All the animal experiments were performed according to the IACUUC protocol number 350, approved by the local committee of IBCh RAS on 21 June 2020.

### 2.2. Immunization and Sample Collection

Before the experiments, the mice were randomly divided into experimental groups in such a manner that in each group, the average weight of the animals (which is one of the main parameters associated with their physiological state) did not significantly differ from the weights in the other groups and was 20–22 g. Only mice without any signs of illness that weighed more than 19 g were included in the analysis. If the animals showed signs of illness or significant depletion of the organism under the influence of the input components (exhaustion, poor condition of the coat, etc.), they were to be withdrawn from the experiment and analysis. However, this was not observed in our work. All the mice used in the experiments had no previous contact with ovalbumin (OVA) or DEPs.

The overall study design is shown in Figure 1. The mice (*n* = 11–12 per group) were immunized for 8 consecutive weeks. They received saline as a control or ovalbumin (OVA; Sigma Aldrich, Darmstadt, Germany) at low (0.3 µg) or high (30 µg) doses, alone or with the diesel exhaust particles (DEPs) SRM1650B (DEP1) or SRM2786 (DEP2) (NIST, USA) 3 times a week for the first 3 weeks and 2 times a week for the remaining 6 weeks. Low (30 µg/mouse) or high (150 µg/mouse) doses of DEPs were used. After immunization with different OVA and DEP doses, some mice (*n* = 5, randomly selected from each group) were challenged with 250 µg of OVA 3 times a week for 2 weeks to evaluate the intensity of allergic lung inflammation in the different groups.

The amount of DEPs administered was established according to previously published doses [19,23]. We administered DEPs only with OVA in the respective groups. The rate of OVA administration was chosen on the basis of preliminary experiments and was similar to that used in our previous allergic model [33]. However, in this model, we used low and high OVA doses that were 3 times higher than those administered by the subcutaneous route in [33] (0.3 µg instead of 0.1 µg and 30 µg instead of 10 µg) because the amount of antigen that reaches the lungs after intranasal administration is 2–3 times less than that administered [34]. Before use, the DEP samples were sonicated for 15 min in saline with 2% normal mouse serum. Immunization was performed by the i.n. route under isoflurane (Baxter) anesthesia in 50 µL of saline containing 2% normal mouse serum that had been taken from the same mice before the experiment. Normal mouse serum was added to the immunization solutions to prevent the DEPs’ aggregation [22].

After 8 weeks of immunization, blood was taken from the suborbital sinus to estimate the specific antibody titers (6–7 serum samples each corresponded to an individual animal from each group were analyzed). The blood samples were incubated for 20 min at 37 °C, followed by centrifugation at 600× *g* to obtain serum samples, which were stored at −20 °C prior to use. For the ELISA experiments, six or seven mice that matched the inclusion criteria from three independent experiments were taken for general analysis.

The next day, the mice were anesthetized with isoflurane and sacrificed by cervical dislocation after the manipulations; the lung tissue and regional lymph nodes were collected and homogenized in ExtractRNA to obtain samples for gene expression measurement. Six samples per group were used for the analysis of gene and transcript expression. For CXCL13 quantification by ELISA, lungs were homogenized in a phosphate saline buffer (PBS) containing 1% Triton-X100. For H&E histology, the lungs were filled with 4% PFA via the trachea, dissected from the thorax, and placed into 4% paraformaldehyde (PFA). For the qPCR experiments, 6 mice (*n* = 6) that match the inclusion criteria from 3 independent experiments were taken for general analysis. For H&E histology, *n* = 6.

To estimate the intensity of allergic inflammation, the mice were euthanized, and bronchoalveolar lavage fluid was taken as described previously by Hoecke et al. [35]. Briefly, bronchoalveolar lavage (BAL) was collected by applying 0.8 mL of ice-cold PBS using an 18 G cannula (Abbocath, ICU Medical, San Clemente, CA, USA) through the trachea to the airways twice. The total BAL volume was approximately 1.3 mL. The BALs were centrifuged at 600× *g*, and the cell pellets were resuspended in 0.2 mL of PBS. For the allergic inflammation measurement, *n* = 5 mice were taken in general analysis.

### 2.3. ELISA

To estimate the specific IgE, IgG1, IgG2a, and IgA production, we performed ELISA as described previously [33]. Briefly, the ELISA for the detection of specific IgE production was carried out in 96-well microtiter plates (Costar, Thermo Scientific, Waltham, MA, USA) coated with 50 mL of a 20 µg/mL OVA solution in PBS (pH = 7.2) overnight at 4 °C. Between each stage, the plates were washed 3–4 times with 0.05% Tween-20 (PBS-T) by using an automatic washer, PP2 428 (Immedtech, Dubna, Russia). After overnight incubation and subsequent washing, the plates were blocked with 5% BSA in PBS for 1 h at room temperature (100 µL in each well). Next, the plates were incubated with different serum dilutions in the same blocking buffer overnight at 4 °C (50 µL in each well). After subsequent washing, we used direct HRP-labeled anti-mouse IgE antibodies. To detect specific IgE, we used anti-mouse IgE-HRP (clone 23G3) at a 1:1000 dilution in a blocking buffer (50 µL in each well). The plates were incubated with this conjugate for 3 h at room temperature. They were then washed again with PBS-T, after which 50 µL of the highly sensitive 3,3′5,5′-tetramethylbenzidine (TMB) substrate (ab 171523, Abcam, Cambridge, UK) was added to each well, before incubation for 30 min. The optical densities (ODs) were measured using an automatic plate reader (Thermo Fisher Scientific, Waltham, MA, USA) at 450 nm, with the subtraction of the optical density at 620 nm. The antibody quantities were estimated as the serum titers corresponding to the maximal serum dilutions at which the ODs were three standard deviations higher than the mean background ODs.

The ELISA protocol for specific IgG1, IgG2a, and IgA production was slightly modified. Coating was performed using a 5 µg/mL OVA solution in PBS (pH = 7.2). Blocking was performed using 1% BSA in PBS. The serum samples and conjugates were also added in 1% BSA in PBS. Following the incubation of the serum samples, to detect specific IgGa, IgG2a, or IgA production, the plates were further processed with anti-mouse IgG1 (clone RMG1-1), anti-mouse IgG2a (clone RMG2a-62), or anti-mouse IgA (clone RMA-1) labeled with biotin, respectively, at a 1:1000 dilution (BioLegend, San Diego, CA, USA) for 2 h at room temperature. After 2 h of incubation at room temperature with primary antibodies, incubation with streptavidin–HRP (BioLegend) at a 1:7000 dilution was performed for 1 h.

For the total IgE measurement, plates were coated overnight with unlabeled anti-mouse IgE (clone RME-1, BioLegend) at a 1 µg/mL concentration. Blocking with 1% BSA in PBS was performed. After that, the plates were incubated overnight with serum samples at different dilutions and processed with biotin-labeled anti-mouse IgEa (clone UH297, BioLegend). Incubation with a biotin-labeled conjugate was performed at room temperature for 3 h. In the final stage, incubation with streptavidin–HRP (BioLegend) at a 1:7000 dilution was performed for 1 h.

The CXCL13 (BCA1) concentration in the lung homogenates was measured using a mouse BCA1 matched antibody pair kit (Abcam, ab218172) according to the manufacturer’s instructions.

### 2.4. Gene Expression Measurements

RNA was extracted using the standard phenol–chloroform method, followed by RNAse-free DNAse treatment (Thermo Fisher Scientific). Briefly, small patches of lung tissue (about 3 mm × 3 mm × 2 mm) from each mouse were homogenized in 1 mL of commercial ExtractRNA solution (Evrogen, Moscow, Russia) containing phenol. For the lymph nodes, 3 × 10^5^ cells in 20 µL of PBS were added to 1 mL of ExtractRNA. The samples were incubated for 30 min at room temperature and then kept at −20 °C prior to RNA purification. 0.25 mL of chloroform was added to 1 mL of the sample during the RNA extraction. The samples were vortexed for 3 min at room temperature, after which they were centrifuged at 15,000× *g* for 15 min at 4 °C. The upper aqueous phase was transferred to a new 1.5 mL tube, and then, 600 µL of isopropanol was added to the sample. The samples were incubated for 30 min at −20 °C for precipitation and then centrifuged for 10 min at 12,000× *g*. After decanting the supernatant, 1 mL of 75% ethanol was carefully added to each sample, after which all the samples were centrifuged again for 10 min at 12,000× *g*. The supernatant was discarded, and the pellets were washed with ethanol again. After that, the samples were heated at 50 °C in uncapped 1.5 mL tubes for 3–5 min to dry up the remaining ethanol.

To remove genomic DNA from our samples, we used RNAse-free DNAse I (EN 0525, Thermo Scientific, Waltham, MA, USA) according to the manufacturer’s protocol. The samples were diluted in 8 µL of DEPC-treated water with the addition of 1 µL of 10× reaction buffer and 1 µL of DNAse I (1U) solution. Incubation at 37 °C for 30 min was performed. To activate DNAse I without RNA degradation occurring in the presence of divalent cations, we added 1 µL of 50 mM EDTA and incubated the samples at 65 °C for 10 min. The remaining template was diluted in a 300 µL volume with DEPC water and kept at −20 °C before usage.

For the measurement of DNA excision circles corresponding to direct and sequential IgE switches, we did not perform DNA digestion; rather, cDNA was synthesized using a RevertAid First Strand cDNA Synthesis Kit (Thermo Fisher Scientific).

Quantitative PCR (qPCR) was performed as previously described [33]. Kits from BioLabMix (Novosibirsk, Russia) were used. We used probes with 6-FAM as a fluorescent dye on the 5′-end and BHQ-1 as a quencher on the 3′-end. The expression of target genes and presence of excision DNA circles were estimated by normalizing to the expression of two housekeeping genes, *GAPDH* and *HPRT*, and calculated as 2^−^^Δ(^^ΔCt)^ compared to the expression in the tissues of intact mice. The data are shown as the relative expression, determined as the ratio of the 2^−^^Δ(^^ΔCt)^ for the experimental groups to that for the intact mice. The reaction was performed in a CFX Connect Amplificator (BioRad, Hercules, CA, USA) according to the following protocol: 95 °C initial denaturation for 3 min followed by 50 cycles of 5 s denaturation at 95 °C and 20 s annealing and elongation at 64 °C. The reaction was performed in 96-well plates (MLP9601, BioRad) in a 20 µL volume. The forward and reverse primer concentrations were both 0.4 µM, and the probe concentration was 0.2 µM. 3 µL of diluted template was used in each reaction. The primers and probes were designed using NIH Primer BLAST and synthesized with Evrogen (Moscow, Russia). The primer and probe sequences used in the study are shown in Appendix A.

### 2.5. Flow Cytometry

The manifestation of allergic airway inflammation was estimated by the presence of myeloid cells in the BAL. For this, mice (*n* = 5 from each group) were randomly selected. BAL fluid was collected as described in Section 2.1. Before cell staining, the cell concentration in the BAL was quantified under a light microscope. Cells in 0.2 mL of PBS were stained with the following anti-mouse antibodies from BioLegend: SiglecF-BV421 (clone 17007L), CD11c-FITC (clone N418), F4/80-PE (clone BM8), Gr-1-PerCP (clone RB6-8C5), CD11b-PECy7 (clone M1/70), and MHCII (I-A/I-E)-APC (clone M5.114.15.2). The incubation was performed for 1 h at 4 °C. Live/dead cell discrimination was performed using a Zombie Aqua (BioLegend) according to the manufacturer’s instructions. The analysis was performed on a MACS Quant Tyto Cytometer (Miltenyi Biotech, Bergisch Gladbach, Germany). The cell concentration in the BAL was determined by the multiplication of the percent of the cell population of interest among the live cells by the total cell concentration in the sample.

The number of eosinophils was estimated as the number of SiglecF+CD11c- cells, that of alveolar macrophages as that of SiglecF+CD11c+F4/80+ cells, and that of neutrophils as that of SiglecF-CD11c-CD11b+Gr-1+ cells, according to slightly modified protocols used by other research groups [35,36]. Our modification of these protocols was the inclusion of an additional marker, SiglecF, as a gating strategy for eosinophil identification. Although in [35], this marker was used for detecting alveolar macrophages, it is also a marker of murine eosinophils [37], which, in contrast to alveolar macrophages, are CD11c^-/low^ [35]. The overall gating strategy is shown in Appendix A.

### 2.6. Histological Analysis of Lung Tissue

Lung tissue samples were subjected to H&E histology. A separate experiment was carried out to obtain samples for H&E histology (*n* = 6–7 for group). The samples were fixed with 4% PFA and kept in it before the preparation of the histological sections. The tissue samples were dehydrated in solutions with increasing ethanol concentrations (70%, 80%, 95%, and 100%) for 45 min and then kept in xylene for 1 h. The tissue samples were then embedded in paraffin. Histological sections 8 nm thick were made on a microtome (Thermo Fisher Scientific HM 355S). The histological sections were rehydrated by incubating them twice for 5 min in a xylene solution and then twice in 100% ethanol for 5 min, followed by two incubations each in 95% ethanol and 70% ethanol for 5 min and, finally, two incubations in water. After filling them with 4% PFA and dissecting them from the thorax, the lung samples were cut at −20 °C on 20 µm-thick sections and were stained using an H&E staining kit (ab245880, Abcam) according to the manufacturer’s instructions. Microscopic images with different magnifications were obtained. In order to estimate the impact of i.n.-administered OVA and DEPs on inducible bronchus-associated lymphoid tissue (iBALT) induction and growth, we quantified the relative area as a ratio: Total area of all histological features/Total area of iBALT structures occupied by iBALTs in these sections.

### 2.7. Statistics

The group means and standard deviations were quantified in each case. ANOVA with correction for multiple comparisons was used to evaluate the significance of the differences between groups. The normality of the distribution of the data in each group was assessed using a Shapiro–Wilk test, and the probability of a normal distribution for each group parameter included in the final analysis was >0.95. The ELISA, qPCR, and BAL cell analysis experiments were performed independently three times. The H&E histological experiments were performed twice. Differences with *p* < 0.05 were considered significant. Only mice that matched the inclusion criteria (normal health status, weight no less than 19 g) were included in the final analysis from independent experiments (*n* = 6–7 for ELISA, qPCR and H&E histology experiments, *n* = 5 for BAL analysis, *n* = 3–5 for chemokine concentration measurement).

## 3. Results

### 3.1. DEPs Induce Formation of Pro-Allergic Antibodies

To evaluate the effects of DEPs on pro-allergic antibody response, we first measured the production of allergen-specific antibodies in mice. Our results show that both DEPs induced IgE-antibody formation at comparable levels when administered at high but not at low doses, so the ability of DEPs to induce IgE responses was relatively independent of the co-administered OVA doses. High doses of DEPs in combination with low (Figure 2A,B) or high OVA doses (Appendix A), in comparison with OVA alone, induced the formation of specific and total IgE in mice after eight weeks of immunization. Low doses of DEPs did not induce IgE-antibody formation in comparison with either saline or OVA alone. 

The DEPs also induced specific IgG1 formation in mice when co-administered with a low (0.3 µg) OVA dose, and the titer of the specific IgG1 was higher upon co-administration with the high (30 µg) OVA dose (Figure 2C and Appendix A). In the case of low OVA doses, both DEP1 and DEP2 at both doses stimulated IgG1 production (Figure 2C). In combination with high OVA doses, only high DEP1 and DEP2 doses stimulated specific IgG1 production (Appendix A). We also measured the levels of specific IgG2a associated with type-1 immune responses [38] and IgA associated with Treg activity in immunized mice [39]. Both DEP1 and DEP2 doses promoted IgG2a formation upon the administration of a high OVA dose (Appendix A). Only high doses of DEPs were able to promote IgG2a production upon immunization with a low OVA dose (Figure 2D). On the contrary, the effects of particles on IgA production were remarkably different. The induction of specific IgA production was observed only upon the administration of DEP2 but not DEP1, together with a high OVA dose. Both low and high doses of DEP2 had statistically significant effects on the IgA titers (Appendix A).

High but not low doses of both types of DEPs stimulated total IgE production in combination with low and high OVA doses. Although the induction of total IgE when DEPs were administered without OVA, compared with the level in saline-administered mice, was not very high, it was statistically significant (Figure 2B and Appendix A). The overall effect of DEPs on antibody production is summarized in Table 1. 

### 3.2. DEPs Stimulate Allergic Inflammation

We next evaluated whether DEP-mediated OVA-specific humoral immune responses resulted in OVA-specific sensitization and subsequent asthma development at an OVA-challenging dose. We focused on the effects of a DEP dose of 150 µg/mouse, since only this particular amount stimulated the formation of pro-allergic antibodies. Despite the comparable levels of specific IgE induced by both high and low OVA doses co-administered together with DEPs, significantly higher titers of IgG1 and IgG2a that accompanied IgE production were observed at high OVA doses (Figure 2 and Appendix A). In steady state, the allergen-specific IgE antibodies are bound to FcεRI receptors on the surface mast cell, and allergen-induced FceRI ligation initiates proinflammatory signaling cascades and the release of anaphylactic mediators. Meanwhile, IgG1 and IgG2a mediate platelet activation factor (PAF)-dependent processes in macrophages [40].

The administration of a challenging OVA dose appears to promote a multicellular immune response (the flow cytometry gating strategy is shown in Appendix A, and representative contour plots are shown in Appendix A). The effects of DEP1 on OVA-dependent neutrophil accumulation in BAL were comparable for mice pre-immunized with low and high doses of OVA. DEP1 induced neutrophils’ accumulation in combination with OVA, in comparison with those in mice immunized with OVA alone. Both low and high OVA doses induced the marginal accumulation of neutrophils in BAL after a high-OVA-dose challenge; however, this effect was not significant, and their accumulation without DEP1 was 3- to 10-fold lower than that with DEP1 (Figure 3B,E). The accumulation of eosinophils (gating strategy shown in Appendix A) in response to DEP1 instillation was observed in the case of their combination with both low and high OVA doses (Figure 3A,D). It should be mentioned that in the absence of OVA, DEP1 was only able to induce insignificant accumulation of neutrophils and eosinophils in BAL after a high-OVA-dose challenge. The alveolar macrophage content in BALs was 2–3 times higher in the high- versus low-dose groups immunized in combination with DEPs after a short high-dose challenge and was also enhanced by DEPs (Figure 3C,F). It is also interesting that long-term DEP administration alone potentiated pronounced macrophage accumulation upon short high-dose challenge, in contrast to the situation with neutrophils and eosinophils. High OVA doses, but not low ones, during eight weeks of immunization further potentiated macrophage accumulation in response to DEPs. In the absence of the particles, a low dose of OVA potentiated macrophages’ accumulation, and a high dose potentiated neutrophils’ accumulation (Figure 3C,E). The data for DEP2 were similar. In summary, for the same DEPs, the allergic inflammation in the low- and high-OVA-dose groups was different. Compared with high doses, low OVA doses induced an allergic immune response that was more dependent on eosinophils and less so on macrophages than in the high-dose groups, although these differences were not crucial, and the number of macrophages usually exceeded the number of neutrophils and eosinophils in each case.

According to our gating strategy, interstitial macrophages, monocytes, and dendritic cells were also present in BAL in negligible numbers.

The impact of DEPs on cellular accumulation in BAL is also summarized in Table 1.

### 3.3. DEPs Induced Local and System Ig Class Switching

To determine whether and where exactly the B-cell class switch occurs, we measured the expression of *germline ε* and *germline γ1* transcripts, as well as circular transcripts corresponding to the direct and sequential class switch to IgE [41]. Because of their rapid degradation in living proliferating cells, DNA excision circles may serve as indicators of sites where the B-cell class switch occurs. High OVA doses were able to induce IgE and IgG1 class switching in lung tissue, but only IgG1 class switching in regional lymph nodes (Figure 4A,B,E,F). However, in the case of lung tissue, the accumulation of transcripts corresponding to mature Ig-producing cells in comparison with that in saline-immunized mice was insignificant (Figure 4C,D,G,H).

Low OVA doses led to the accumulation of IgG1 class switch marker (*germline γ1*) only but not mature IgG1 producing cells markers (*postswitch γ1*) in both the lungs and the regional lymph nodes (Figure 4B,F). In either case, no statistically significant accumulation of mature Ig transcripts was detected in mice immunized with 0.3 µg of OVA in comparison with a saline control. Despite the fact that both types of particles induced IgE production at comparable levels, their properties in relation to the stimulation of local or systemic isotype switching were slightly different. DEP1 induced both local (in lung tissue) and systemic (in regional lymph nodes) isotype switching to IgE (*germline*
*ε*) when administered with low OVA doses (Figure 4A,E).

DEP1 administered with high OVA doses stimulated only local IgE class switching (Figure 4A). Only local IgE class switching was induced by DEPs, independent of their type, with both low and high OVA doses (Figure 4A). DEP1 administered in the absence of OVA induced only local IgE class switching, and DEP2 in the absence of OVA induced only systemic IgE class switching (Figure 4A,E). DEP2 induced both local and systemic IgE class switching in combination with both OVA doses in comparison with mice immunized with OVA alone (Figure 4A,E). In combination with low OVA doses, DEPs induced local but not systemic IgG1 class switching (Figure 4B,F). In combination with high OVA doses, DEP2 induced IgG1 class switching at both sites, while DEP1 induced only local IgG1 class switching in the lungs. Therefore, both types of particles accelerated the IgG1 class switching (*germline γ1* expression) in lung tissue after the administration of low OVA doses (Figure 4B). However, in the absence of OVA, both types of particles were capable of inducing *germline γ1* expression at both sites. Both types of particles induced the accumulation of *postswitch ε* transcripts in the lungs and regional lymph nodes, as well as DEP2 in combination with low OVA doses in the case of the lymph nodes (Figure 4C,G). 

Therefore, in the lung tissue, the IgE class switching (*germline ε* transcripts) and IgE production (*postswitch ε* transcripts) induced in response to DEPs in the OVA-immunized animals were more independent on the administered OVA dose and type of DEP. In the DEP1-treated low-OVA dose group, both direct (µ–ε) and sequential (γ1–ε) mechanisms mediated the local IgE class switch (Appendix A). However, only direct IgE class switching was observed in the high- OVA dose group (Appendix A). In the lymph nodes, DEP1 and low OVA doses stimulated B-cell IgE class switching only by the direct mechanism (Appendix A). DEP2 induced both local and systemic IgE class switching in both the low- and high-OVA dose groups (Figure 4A,E). Similarly to DEP1, DEP2 significantly induced local direct and sequential IgE class switching in the low-dose group (Appendix A). Both IgE-switching mechanisms were induced in regional lymph nodes after DEP2 administration with low OVA doses (Appendix A). In the high-OVA group, DEP1 but not DEP2 exerted positive effects on IgG1-producing cell accumulation, and only in the lungs but not in the lymph nodes (Figure 4D,F). It is interesting that DEP1 even reduced the accumulation of *postswitch γ1* transcripts in regional lymph nodes in the high-OVA-dose group. We do not exclude the possibility that during long-term immunization under conventional conditions, certain minor irrelevant environmental antigens could enter the lung tissue along with the experimentally administered DEPs. This may explain the expression of certain markers of B-cell class switching and Ig production even in the absence of OVA (Figure 4). The effects of high DEP1 and DEP2 doses on antibody class switching and Ig production markers’ expression in the lungs and regional lymph nodes are summarized in Table 2.

### 3.4. Histological Analysis of iBALTs in the Lungs of Mice after OVA and/or DEP Administration

Due to the lung localization of both the DEPs and OVA-mediated dose-independent B-cell switching and accumulation of IgE-producing cells, we next focused on local B-cell class switching in detail. The local immune response usually depends on tertiary lymphoid structures such as iBALTs in the lungs [42]. Therefore, we decided to examine whether iBALT development and growth in lung tissue were accelerated by DEPs. Despite the fact that both types of DEPs at high doses induced local IgE class switching and IgE production when administered with low OVA doses, only high doses of DEP2 induced iBALT formation per se and in combination with 0.3 µg of OVA (Figure 5A, Table 1). Both types of DEPs, however, within the entire range of concentrations, stimulated iBALT formation in response to the high dose of OVA (Figure 5B, Table 1). Despite the fact that (1) low OVA doses in combination with high DEP2 doses potentiated iBALT formation and growth in the lungs (Figure 4a) and (2) high OVA doses administered with both doses of DEP1 and a high dose of DEP2 triggered iBALT formation (Figure 5B, Table 1), low OVA doses in combination with a high DEP1 dose induced IgE production (Figure 1) but did not trigger iBALT formation (Figure 5A,B, Table 1). From these facts, it becomes clear that in some groups where iBALT formation and growth were significantly triggered, local IgE production was absent or insignificant. 

On the other hand, in the group immunized with a low OVA dose and high DEP1 dose, local B-cell IgE class switching was induced, but significant iBALT growth was not detected. Representative histological images of the lung tissues show that the sites of DEP accumulation, especially for DEP1, which was prone to forming larger aggregates, did not always coincide with either the sites of de novo iBALT formation or the sites of immune cell infiltrates (Figure 5C and Appendix A).

### 3.5. Stimulation of Cytokines and CXCL13 Chemokine Expression in Response to OVA and/or DEP Administration

It is curious why two closely related DEPs exerted different effects on iBALT formation while stimulating similar pro-allergic humoral responses in terms of antibody production and the induction of local Ig class switching. It is known that CXCL13, produced by stromal cells in response to TNFα- and LTα-based interactions with B cells, is crucially involved in the initiation of the orchestration of lymphoid tissue’s remodeling and lymphoid neogenesis [42,43,44]. To evaluate the ability of two types of DEPs to stimulate CXCL13 production, we measured the levels of chemokine production in the lung homogenates of immunized mice. Indeed, in contrast to DEP1, DEP2 in combination with a low OVA dose weakly but significantly triggered CXCL13 production. DEP1 mediated the increase in CXCL13 production only in the high-OVA-dose group (Figure 6A). It is also important to note that the OVA alone, at either a low or high dose, was not able to stimulate CXCL13 production (Figure 6A), which is in agreement with data regarding the absence of significant de novo iBALT formation in response to the OVA alone (Figure 5).

It is unlikely that the different abilities of DEP1 and DEP2 to potentiate iBALT formation and growth in lung tissue in combination with low OVA doses are linked to different abilities to induce the expression of iBALT-promoting proinflammatory cytokines. It is generally accepted that tumor necrosis factor α (TNFα) contributes to iBALT formation [43]. High OVA doses induced *TNFα* gene expression in lung tissue without DEPs and also with both types of DEPs. In the low-OVA-dose group, the induction of *TNFα* gene expression was only observed upon treatment with DEPs, regardless of their type, but only DEP2 was able to stimulate its expression in the high-OVA-dose group (Figure 6B). It was recently shown that type-I interferons are also able to induce iBALT formation [45]. Both types of DEPs induced *interferon α1* (*IFNα1*) expression independently of the co-administered OVA dose. OVA alone did not induce *IFNα1* expression, at either a low or high dose (Figure 6C).

Local IgE class switching and IgE production must depend on the production of certain cytokines, particularly IL-4 [1], regardless of the presence or absence of iBALT formation. Indeed, both types of DEPs stimulated *IL-4* expression in lung tissue independently of the OVA dose (Figure 6D). Although the expression of *IL-13* in the mice immunized with DEPs in combination with OVA was not significantly higher than that in the saline-immunized mice, it was significantly higher than that in the mice immunized with OVA alone, irrespective of the DEP type and OVA dose (Figure 6E). At the same time, we could not detect the induction of *IFNγ* expression by DEPs in the mouse lungs; moreover, in the case of a low OVA dose, DEP administration repressed its expression (Figure 6F). Therefore, both types of DEPs stimulated the local production of cytokines associated with B-cell IgE class switching.

Despite the different impacts of the two types of DEPs on iBALT formation, both induced B-cell accumulation in the lungs, as can be judged by *Cd19* expression, and this effect was slightly dependent on the OVA dose (Figure 7A). High OVA doses also induced germinal centers’ formation in lung tissue (marked by *Bcl6* expression, which is a germinal center marker gene [46]). DEP1 did not stimulate their formation in the low-OVA-dose group. DEP2, rather than DEP1, in combination with low OVA doses, stimulated germinal centers’ formation in the lungs (Figure 7B). 

Both types of particles triggered the expression of *Ebi2*, a marker of extrafollicular focus formation [46], and this effect was independent of the co-administered OVA dose (Figure 7C). The effects of high DEP1 and DEP2 doses on CXCL13 production and proinflammatory cytokine genes’ expression in the lungs are shown in Table 3.

## 4. Discussion

It is generally accepted that DEPs induce the formation of pro-allergic antibodies and the subsequent development of asthma in humans and in laboratory animal asthma models [5,6,16,17,18,19,20,21,22,23]. However, in most of the research, only one type of particle was used. It is known that the ability of particulate matter to enter the lungs and to trigger local and systemic inflammation depends on the size of the particles. In our work, we used two types of DEPs: DEP1 and DEP2. The former originate directly from the heat fuel exchangers of diesel engines; they are more polydisperse in nature and tend to form large aggregates despite their small size [21]. The latter were collected from the air [32] and may represent a mixture of particles derived from industrial sources and from incomplete fuel combustion. On the other hand, they may represent a fraction of the former that were not prone to aggregation. Therefore, they remained in the air for longer.

Despite such differences, the particles stimulated humoral immune responses at comparable levels. Although only DEP2 stimulated IgA formation, the resulting IgA titers were not very high in comparison with the IgG titers, and this phenomenon may not have very marked effects in relation to allergic inflammation. In relation to IgE, the effects of these particles on IgG1 and even IgG2a production were similar. While, in some cases, IgG1 and IgG2a production were stimulated by relatively low DEP doses, IgE production was only stimulated by high DEP doses (immunization with 150 µg/mouse, or 15 mg/kg weekly). Although such amounts of DEPs usually do not penetrate into the human body, even in heavily polluted industrial or urban areas, it should be noted that DEPs tend to accumulate in the body over a long time and, therefore, reach high doses [2,3]. Our data are in agreement with research indicating that only high DEP doses potentiate allergic inflammation [21], contradicting others in which even low DEP doses triggered type-2 immune responses [23].

However, in the latter study, the allergen extract contained additional stimuli as opposed to pure protein [23]. The fact that only high DEP doses induce IgE production and asthma development may indicate that such substances account for asthma and allergy development per se only in regions with heavy industrial and road traffic burdens. Although in most European and North American countries, fuel and engine standards presume very low amounts, if any, of particulate matter after combustion compared with 30–35 years ago [47], one can conclude that soon, this type of air pollution will cease to be the leading cause of the increasing asthma prevalence. Unfortunately, in developing countries in Asia and Africa where diesel fuel is widely used, as well as fuels based on coal and wood, and where environmental regulations are not so stringent, this problem is very serious [48].

We previously showed that upon the administration of antigen in fat tissue with tertiary lymphoid structures, mainly low but not high antigen doses induced IgE production [33]. However, the immune response upon the administration of antigen in lung tissue may be substantially different from that initiated after administration in subcutaneous fat due to the different properties of either tertiary lymphoid structures in lungs and fat tissue or different rates and methods of antigen delivery to secondary lymphoid organs from these two sites. Indeed, in the present work, DEPs stimulated IgE synthesis in response to both low and high OVA doses. This fact indirectly shows that the impact of DEPs on IgE responses is not linked to the induction of germinal centers because these structures develop poorly when the OVA is administered at low doses. This was confirmed by the absence of any significant induction of the germinal center marker *Bcl6* in mice immunized with low doses alone or with low OVA doses with DEP1. Therefore, DEPs stimulate an OVA-specific pro-allergic humoral response over a wide range of OVA doses, in contrast to the situation observed in the adjuvant-free model in our previous work [33].

Generally, DEPs stimulate the accumulation of three major types of inflammatory cells in BAL fluid in our model: eosinophils, macrophages, and neutrophils, as well as OVA-specific IgE and type-2 cytokines. Therefore, our DEP-induced asthma model is appropriate for the evaluation of the effects of potential therapeutics on the key parameters of allergic asthma linked to the type-2 immune response. 

In our asthma model, DEPs potentiated eosinophils’ accumulation after high-dose-OVA challenge when combined with OVA during eight weeks of immunization. The number of eosinophils in BAL was slightly higher in the mice immunized with a low (0.3 µg) OVA dose in combination with DEP1 than in the mice immunized with a high (30 µg) OVA dose, but the difference was not significant. On the other hand, high but not low OVA doses synergized with DEPs in the potentiation of macrophages’ accumulation. Generally, the number of macrophages was significantly higher than the number of eosinophils and neutrophils. All the particles induced significant expression of type-2 cytokine genes in the lungs. Therefore, we conclude that in our case, DEPs induced T-helper-2-type-associated asthma involving eosinophils, IL-4, and IL-13. We used high DEP doses in this study, which, according to [9], are more prone to inducing T-helper-17- and neutrophil-associated inflammation. However, we administered DEPs 2–3 times a week, in contrast with some studies in which DEPs were administered 5 times a week, and this could explain the differences. Our results regarding the accumulation of eosinophils and neutrophils in response to DEPs are in agreement with some literature data in that the administration of this type of particle also potentiated the accumulation of eosinophils and (to a lesser extent) neutrophils in BAL [19,23], but this contradicts other studies, in which only neutrophil accumulation was observed [21]. However, in [21], DEPs were administered without antigen, and in our work, the induction of eosinophil accumulation was also observed, mostly in mice immunized with a combination of DEPs and antigen. The accumulation of macrophages in BAL was not studied in some previous work [9,23] and was not observed in response to DEPs in [21,22]. However, the authors of [21] used DEPs generated in their laboratory as opposed to those from NIST standard samples, and in [22], only a single DEP dose was administered to mice. This may be the reason for the different results obtained by these groups. In this study, we have shown that DEPs induced significant macrophage accumulation in BAL; these macrophages were CD11c+SiglecF+ and thus belonged to the alveolar macrophage subpopulation. This is in agreement with the results obtained for monodisperse particles, showing that such macrophages were the main triggers of airway inflammation and local antibody production [25].

DEPs stimulate both local and systemic IgE class switching when administered with low OVA doses. However, in the case of SRM1650B, we observed only local IgE class switching after their co-administration with a high OVA dose, and in the case of SRM2786, only the accumulation of local IgE-producing cells was observed after their co-administration with a high OVA dose. Despite the fact that only local IgE class switching was induced by DEPs, independent of the DEP type and OVA dose in OVA-immunized animals, *postswitch ε* transcripts accumulate in both the lungs and lymph nodes. This may be due to the migration of antigen-experienced B cells between two sites, as was shown by another research team for lung iBALT [49] and as was postulated in our previous work with subcutaneous-fat-associated structures [33]. 

From our previous work, we concluded that in young allergic patients [50] and in laboratory animals [33], IgE class switching occurs mainly by direct mechanisms. However, the results of this study contradict this, because we observed the accumulation of both types of circular transcripts corresponding to IgE class switching from IgM and from IgG1 as well. This is in agreement with some results that showed that both types of B-cell IgE class switching could be observed in allergic patients [51]. Even in animal models, this switching may occur via either a direct [52] or a sequential mechanism [53], depending on the model used. However, it should be mentioned that when co-administered with high OVA doses, when intensive iBALT growth is observed, DEPs induce only direct IgE class switching. This situation is closely related to clinical cases of patients with asthma or chronic allergic rhinitis in the later stages of disease development when high antigen doses accumulate in iBALT or nasal polyps for a long time [51]. Therefore, direct IgE class switching may be more important, but further studies are needed to answer this question. The lack of a significant effect of DEPs on the expression of *germline* and *postswitch γ1* transcripts in some cases may reflect the fact that in the case of IgG1, DEPs stimulate its production without increasing IgG1 class switching per se but rather by accelerating the differentiation of IgG1-expressing B cells to the final stages of IgG1-producing plasma cells. It is also possible that the effects of DEPs on IgG1 class switching and the accumulation of IgG1-producing cells are stronger and more significant in the early stages of the process.

It is tempting to speculate that the different abilities of the two types of particulate matter to induce iBALT formation and growth after accumulation in the lung tissue are due to their different tendencies to form relatively large (>10 µm) aggregates. This is despite the fact that prior to the administration to mice, when the DEPs were sonicated, lung microphotographs from H&E histology showed that large aggregates of particles were present in the lung tissue, especially in the case of DEP1. According to their characteristics as revealed by NIST, DEP2 particles, despite their larger size prior to aggregation, are less prone to forming large aggregates [32]. Large aggregates slowly diffuse through the epithelium and are hardly phagocytized by APCs. Therefore, despite their ability to damage airway epithelium, they are hardly delivered into the lung tissue with small soluble antigens, and for this reason, their ability to induce iBALT could be limited. On the other hand, SRM2786 particles that have been in the air for a long time may have a different chemical composition from SRM1650B, and this could be responsible for such differences.

DEP2 (SRM2786), in comparison with DEP1 (SRM1650b), was able to stimulate IgA production in mice immunized with high OVA doses. Though IgA class switching is usually associated with Tregs [39], DEP2, in our study, showed more proinflammatory potential than DEP1. Indeed, DEP2, in contrast to DEP1, stimulated *germline ε* and *germline γ1* transcripts’ expression and increased the amount of *circular µ-ε*, not only in the lung tissue but also in the regional lymph nodes in high-OVA-dose-immunized animals. DEP2 was more potent than DEP1 in the stimulation of *Tnfa* expression in the lungs in combination with high OVA doses. Therefore, if some Tregs’ responses were indeed induced by DEP2 in the high-dose-immunized mice, it may be due to the induction of a compensatory response at later stages that is dispensable and insignificant for the inhibition of general inflammation [54]. This seems not to be very important because of the low levels of induced IgA.

A more important observation is that in combination with low OVA doses, DEP2 but not DEP1 induced de novo iBALT formation and growth, despite their similar ability to induce IgE production. This difference is linked to their different abilities to stimulate initial CXCL13 production and germinal center responses when the particles are co-administered with low OVA doses. It is likely that particles from the fine type of particulate matter DEP2 adsorb the co-administered antigen on their surface and then penetrate the epithelial barrier with it. When these particles accumulate in tissue, the adsorbed antigen serves as a stronger signal for B and T cells than soluble antigen [55]. Certainly, DEP1 particles can also adsorb a proportion of OVA on their surfaces, but when they form large aggregates, they cannot pass through the epithelial barrier, and the proportion of antigen adsorbed on their surface remains in the alveoli or bronchial space, out of reach of B and T cells. The other possibility is that specific chemicals present in DEP2 but absent in DEP1 serve as additional triggers for CXCL13 production. It is worth mentioning that DEP1 induced the accumulation of *postswitch ε* and *postswitch γ1* transcripts in both the lungs and lymph nodes in combination with low OVA doses, but DEP2 did not induce the accumulation of the former in the lymph nodes and the latter in lungs. This may indicate that in the case of DEP2, the migration of mature Ig-producing B cells between two sites is more restricted. Both types of particles in combination with low OVA doses induce proinflammatory cytokine expression in lungs at comparable levels, as well as the B-cell activation marker *Ebi2* associated with the extrafollicular B-cell response [46]. However, by contrast, DEP2 but not DEP1 stimulated Bcl6 induction in the lungs, which is associated with germinal centers [56]. However, the two types of particles induced IgE production and local IgE class switching at comparable levels. Therefore, the abovementioned differences are not responsible for their ability to stimulate IgE responses. Both types of DEPs induced local *Ebi2* expression, which is an indicator of extrafollicular focus formation [46]. Both types of DEPs also induced local *TNFα* and *IFNα1* expression, as well as *IL*-4 and *IL*-13, but the expression of *IFNγ* was not induced. From these data, it becomes clear that DEPs induce local IgE production by the specific stimulation of type-2 immune responses; this is in agreement with most previous studies [5,20,21,23]. This common property of the two types of DEPs may be more important than their different effects on iBALT formation for the induction of allergic asthma. Though it was shown that DEPs [22] or their carbon cores [5] are also capable of inducing IFNγ production, we did not observe the induction of *IFNγ* expression in the mouse lungs in our work. As mentioned above, the authors of [22] used a single DEP instillation, so this situation is different from ours, with chronic DEP administration. Despite the fact that the DEPs stimulated IgG2a production in some cases, this may indicate that this effect was due to an increase in proinflammatory cytokine production, which is common for Th-1 and Th-2 immune responses (for example, Tnfa), and not due to an increase in local IFNγ production or Th-1 activation in the lungs. It is also possible that some Th-1 activation and IFNγ production may occur in the lymph nodes but not in the lungs; this was attributed to an increase in IgG2a production.

The main conclusion we can draw from this work is that local IgE production in response to particulate matter is independent of the ability of the particulate matter to induce iBALT formation and growth. According to qPCR data, this local IgE response was also independent of germinal centers; this is in agreement with recent results obtained by other researchers that the local IgE class was mostly associated with the extrafollicular response but not with germinal centers [57,58] and by our group in a model with subcutaneous immunization [59]. The common properties of these two types of DEPs must be responsible for the induction of local B-cell IgE class switching. In the case of the DEP1 induction of *TNFα* and *IFNα1* expression, when these particles were co-administered with low OVA doses, it did not lead to iBALT induction and growth, notwithstanding the fact that these cytokines have iBALT-promoting activity [42,43,45]; instead, in our case, iBALT induction seemed to be linked to CXCL13 production, which is indispensable for this process [42,45]. Our results regarding the absence of the ability of DEP1 to induce iBALT formation in combination with low OVA doses seem to conflict with results obtained by Hiramatsu et al. [26], who found that DEP inhalation induced iBALT formation in mouse lungs. However, these authors used DEPs generated in their laboratory rather than the standard ones obtained from NIST; additionally, in that study, DEPs were inhaled five days a week [26]. In our study, DEPs were administered 2–3 times a week, so the resulting dose in our case could be lower. The situation in [26] is closer to that in this study with respect to DEP2, which, in high doses, was able to induce iBALT formation independent of the OVA dose.

However, from our work, it becomes clear that local IgE production, surprisingly, does not always require de novo iBALT formation and growth, which does not mean that iBALT per se was not required for IgE B-cell class switching. We also cannot exclude the possibility that in the case of more frequent administration of DEPs (5 times a week instead of 2–3 times as in [26]), the resulting local immune response would be different from that observed in our study and the relationship between iBALT growth and IgE production would be stronger. However, the other conclusion that our data allow us to draw seems to be more important, namely, that iBALTs were present even in naïve mice despite their low numbers and small sizes, so it is likely that even small and rare iBALTs are sufficient to spur local specific IgE production. This is because, according to some data, fully mature germinal centers hamper IgE class switching [38] and larger iBALTs contain such germinal centers, so it could be that small iBALTs represent a place where IgE class switching actually occurs. Indeed, according to some data, large iBALTs, if formed prior to allergen administration to mice, do not enhance but can even dampen allergic inflammation as well as total IgE production—though, in that case, iBALT was induced by LPS but not by DEPs [60]. Despite the fact that DEPs do not induce iBALT growth, they induce extrafollicular B-cell activation and CD19-expressing B-cell accumulation in the lungs. One interesting hypothesis is that IgE class switching may occur in B cells situated beyond iBALTs in the lung tissue in B-cell aggregates and infiltrates. B cells entering nonimmune organs do not always form tertiary lymphoid structures, though in such cases, they represent cells recirculating from secondary lymphoid organs to peripheral tissues [61]. Poorly organized B-cell aggregates were also found in peripheral tissues in some cases [62]. A recent study shows that in immunized mice, IgE-producing B-cells formed such aggregates, though they were found in the spleen but not in tissues, and the authors suggest that IgE-producing plasma cells form such aggregates after IgE switching due to the appearance of specific receptors on their surface [63]. However, it is tempting to speculate that specific FcγRII/III on the surfaces of IgE plasma cells, if it does not trigger IgE class switching alone, may serve as a prosurvival signal for recently switched IgE+ B cells because the inhibition of such aggregates’ formation leads to reduced serum IgE levels in mice. If so, stimuli that enhance B-cell accumulation in tissue, and rapid proliferation and differentiation in antibody-producing cells may lead to such aggregates’ formation where B cells receive pro-survival signals even beyond the organized tertiary lymphoid clusters. Further work needs to be conducted to determine whether one of these hypotheses is correct. Due to the paucity of IgE+ cells in the overall B-cell population and the expression of CD23 that binds exogenous IgE from serum [64], it is difficult if not impossible to address this question using conventional immunohistochemistry. However, if IgE+ cells indeed formed locally or in regional lymph nodes but outside the B-cell follicles of these structures, or even outside tissue tertiary lymphoid clusters or outside B-cell zones of lymph nodes, their formation must be insensitive to specific inhibitors of signaling pathways and cytokines responsible for tertiary lymphoid clusters and B-cell follicle formation (for example, being insensitive to Tnfa inhibitors and *Bcl6* inhibitors), and must be sensitive to specific inhibitors of pathways responsible for the formation of B-cell aggregates. Further work is required to clarify this matter.

## 5. Conclusions

Overall, DEPs can induce local as well as systemic B-cell IgE class switching, which leads to allergic immune responses. However, local but not systemic IgE class switching in response to DEPs could be observed independent of the OVA dose and type of particle. This local IgE class switch occurs mainly through direct mechanisms, but low OVA doses can also stimulate a sequential switch. Only high DEP doses induce IgE production, so DEPs may be responsible for atopic disease development only when present constantly and in very high concentrations in the air, e.g., if someone lives in a region with a high road traffic burden, or in allergy-prone individuals. Local DEPs induced B-cell IgE class switching independent of their ability to induce the expression of CXCL13, germinal center development, and de novo iBALT formation and growth. Small iBALTs initially present in the lungs or other structures may be responsible for IgE production in this case. DEP-induced local B-cell IgE class switching was accompanied by extrafollicular B-cell activation and the induction of the expression of cytokines associated with type-2 immune responses in the absence of type-1 immune response activation.

## Figures and Tables

**Figure 1 ijerph-19-13063-f001:**
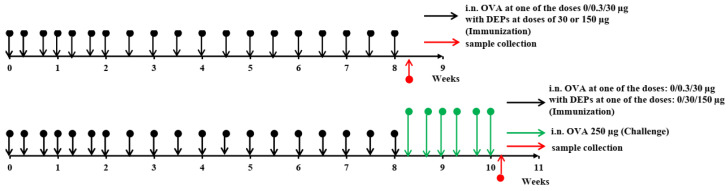
Study design. Black arrows indicate i.n. immunization of mice with OVA alone at a 0.3 µg or 30 µg dose or in combination with DEPs at a low (30 µg) or high (150 µg) dose. Animals immunized with saline alone without OVA were used as a control group. Green arrows indicate a high-dose (250 µg) OVA i.n. challenge. The time points when the mice were killed and when the samples were collected are indicated by red arrows.

**Figure 2 ijerph-19-13063-f002:**
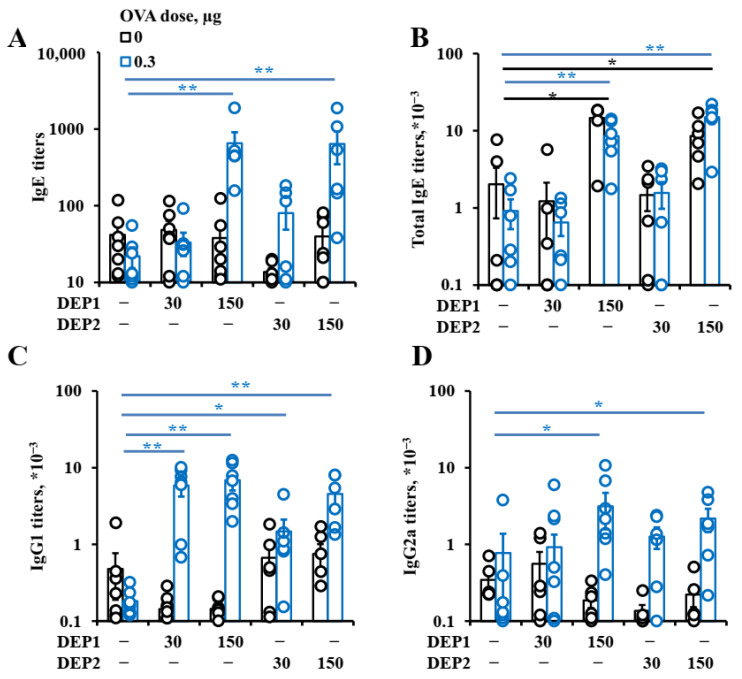
Antibody production in response to low OVA doses, DEPs, or their combination. BALB/c mice were immunized 2–3 times a week for 8 weeks (a total of 18 immunizations) via the i.n. route with 0.3 µg of OVA alone or with different types of DEP at 30 or 150 µg. The titers of OVA-specific IgE (**A**), IgG1 (**C**), IgG2a (**D**), and total IgE (**B**) were measured. The statistically significant differences of the groups immunized with OVA with particles vs. the OVA control are shown with blue bars, and the statistically significant differences of the groups immunized with particles alone vs. the saline control are shown by black bars. (*—*p* < 0.05; **—*p* < 0.01). The experiment was performed three times. The graphs show the data from three representative experiments (total number of mice included in analysis from all experiments *n* = 6–7 per group). The data indicate biological replicates obtained from individual mice.

**Figure 3 ijerph-19-13063-f003:**
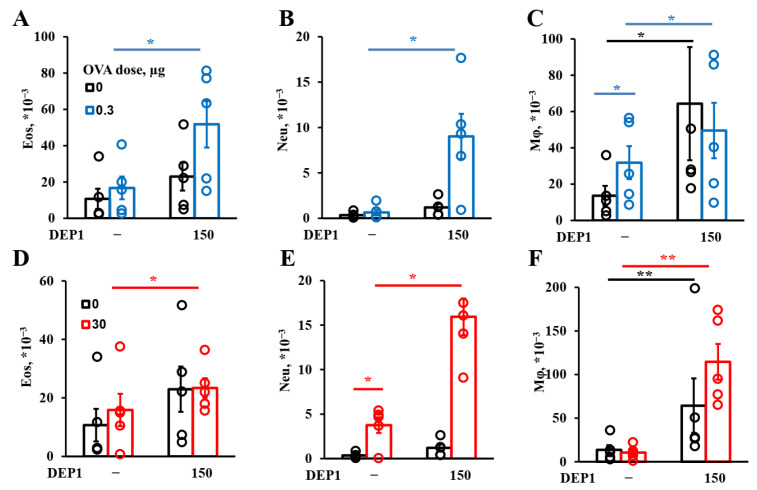
Cell response to DEPs. BALB/c mice were immunized with a low dose of OVA, 0.3 µg (**A**–**C**), or high dose, 30 µg (**D**–**F**), alone or with 150 µg of DEP1. After this, the mice were challenged three times a week for two weeks with 250 µg of OVA and sacrificed. BALs were taken, and the eosinophils (Eo), neutrophils (Neu), and alveolar macrophages (Mϕ) were quantified with flow cytometry. The experiment was performed three times. Representative data are from one out of three independent experiments. Significant differences vs. PBS are shown with black bars; those vs. 0.3 µg of OVA with blue ones; and those vs. 30 µg of OVA with red bars (*—*p* < 0.05; **—*p* < 0.01). The graphs show the data from three representative experiments (total number of mice included in analysis from all experiments *n* = 5 per group). The data indicate biological replicates obtained from individual mice.

**Figure 4 ijerph-19-13063-f004:**
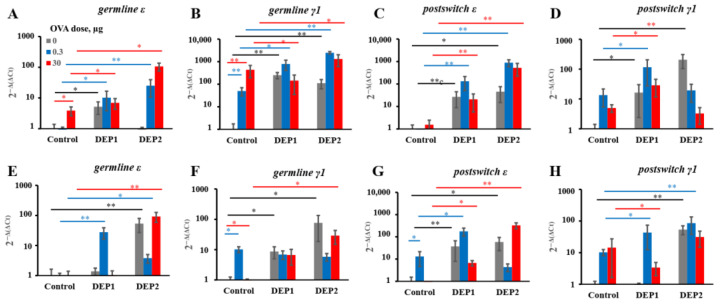
Expression of transcripts corresponding to B-cell Ig class switching and antibody production in lung tissue and regional lymph nodes. BALB/c mice were immunized 2–3 times a week for eight weeks (a total of 18 immunizations) via the i.n. route without OVA (Saline), with a low (0.3 µg) OVA dose or high (30 µg) OVA dose, without particles (control), or with either DEP1 or DEP2 in high (150 µg) dose. After immunization, the mice were sacrificed. The expression of transcripts corresponding to IgE (*germline ε*) (**A**,**E**) and IgG1 (*germline γ1*) (**B**,**F**) class switching, as well as transcripts corresponding to mature IgE (*postswitch ε*) (**C**,**G**)- and IgG1 (*postswitch γ1*) (**D**,**H**)-producing cells were measured in lung tissue (**A**–**D**) and regional lymph nodes (**E**–**H**). Black bars indicate significant (*—*p* < 0.05; **—*p* < 0.01) differences between mice immunized without OVA with different DEPs and the respective control without DEPs; green bars indicate significant differences between mice immunized with low OVA doses with or without DEPs and the respective control groups; red bars indicate significant differences between mice immunized with high OVA doses with or without DEPs and the respective control groups. The experiment was performed three times. The graphs show the data from three representative experiments (total number of mice included in analysis from all experiments *n* = 6 per group). The data indicate biological replicates obtained from individual mice.

**Figure 5 ijerph-19-13063-f005:**
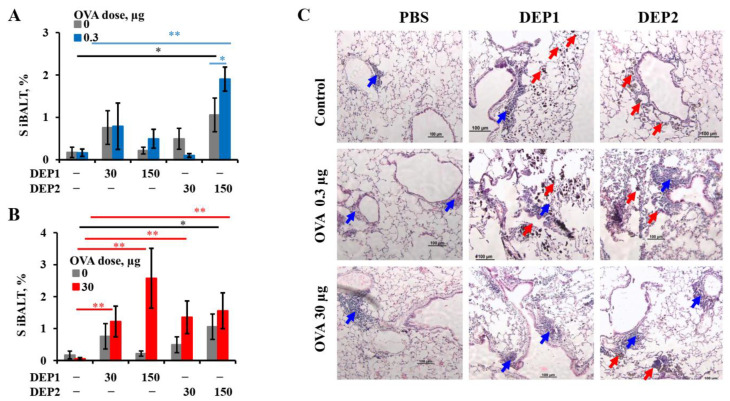
Histological analysis of lung tissue after prolonged OVA and DEP administration. BALB/c mice were immunized 2–3 times a week for eight weeks (a total of 18 immunizations) via the i.n. route with the indicated doses of OVA and different doses of DEP1 or DEP2. Following immunization, the mice were sacrificed and the lungs were isolated and stained for histological analysis (H&E). Relative areas of iBALT on histological sections of mice immunized with PBS and low OVA doses with or without DEPs in either 30 or 150 µg dose (**A**) or with high OVA doses with or without the same doses of DEPs (**B**), and representative histological images (100×) (samples from mice immunized with or without DEPs in 150 µg dose) (**C**) are shown. Red arrows correspond to DEPs; blue arrows show iBALTs. Black bars show significant differences between the groups immunized with particles alone and PBS control (*—*p* < 0.05; **—*p* < 0.01). Blue and red bars show differences between the mice immunized with low (blue) or high (red) OVA doses together with DEPs and mice immunized with OVA alone. The experiment was performed twice. The graphs show the data from three representative experiments (total number of mice included in analysis from all experiments *n* = 6–7 per group). The data indicate biological replicates obtained from individual mice.

**Figure 6 ijerph-19-13063-f006:**
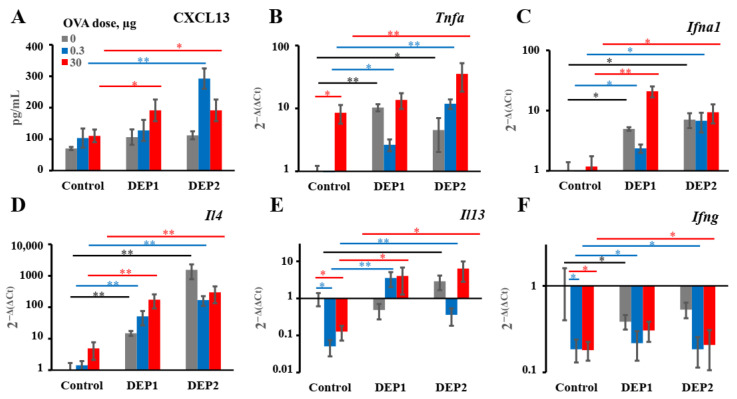
Production of iBALT-inducing chemokines and expression of cytokines in the lung tissue of immunized mice. BALB/c mice were immunized as described in Figure 4 with or without high (150 µg) DEPs doses. Control mice were immunized in the absence of DEPs. The production of CXCL13 (**A**), the expression of the iBALT-promoting cytokines Tnfa (**B**) and Ifna1 (**C**), type-2 immune responses promoting the cytokines Il4 (**D**) and Il13 (**E**), and type-1 immune responses promoting cytokine Ifng (**F**) were measured in the lung tissue. Black bars indicate significant (*—*p* < 0.05; **—*p* < 0.01) differences between the mice immunized without OVA with different DEPs and the respective control without DEPs; green bars indicate significant differences between the mice immunized with a low OVA dose with or without DEPs and the respective control groups; red bars indicate significant differences between mice immunized with a high OVA dose with or without DEPs and respective control groups. The experiment was performed three times. The graphs show the data from three representative experiments (total number of mice included in analysis from all experiments *n* = 6 per group). The data indicate biological replicates obtained from individual mice.

**Figure 7 ijerph-19-13063-f007:**
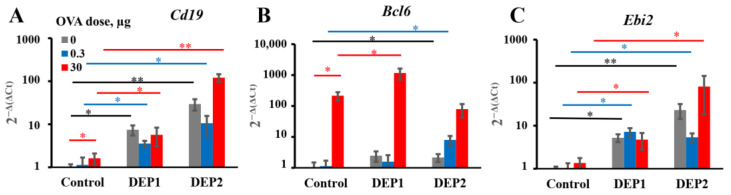
Expression of B-cell-related genes in lung tissue. BALB/c mice were immunized as described in Figure 4 with or without high (150 µg) DEPs doses. Control mice were immunized in the absence of DEPs. Expression of markers for relative B-cell numbers (Cd19; (**A**)), germinal center formation (Bcl6; (**B**)), and B-cell extrafollicular activation (Ebi2; (**C**)) is shown. Black bars indicate significant (*—*p* < 0.05; **—*p* < 0.01) differences between the mice immunized without OVA with different DEPs and the respective control without DEPs; green bars indicate significant differences between the mice immunized with low OVA doses with or without DEPs and the respective control groups; red bars indicate significant differences between the mice immunized with high OVA doses with or without DEPs and the respective control groups. The experiment was performed three times. The graphs show the data from three representative experiments (total number of mice included in analysis from all experiments *n* = 6 per group). The data indicate biological replicates obtained from individual mice.

**Table 1 ijerph-19-13063-t001:** The effect of DEPs on antibody production, cellular composition in BAL, and iBALT growth in immunized mice in comparison with those in mice immunized without DEPs with the indicated OVA doses.

Estimated Parameter	OVA Dose, µg	DEP1, 30 µg	DEP1, 150 µg	DEP2, 30 µg	DEP2, 150 µg
OVA IgE titer	0	0	0	0	0
	0.3	0	++	0	++
	30	0	++	+	+
Total IgE titer	0	0	++	0	+
	0.3	0	++	0	++
	30	0	+	0	++
OVA IgG1 titer	0	0	0	0	0
	0.3	++	++	+	++
	30	0	+	0	+
OVA IgG2a titer	0	0	0	0	0
	0.3	0	+	0	+
	30	+	+	++	+
OVA IgA titer	0	0	0	0	0
	0.3	0	0	0	0
	30	0	0	+	++
Eos	0	ND	0	ND	0
	0.3	ND	+	ND	+
	30	ND	+	ND	+
Neu	0	ND	0	ND	0
	0.3	ND	+	ND	+
	30	ND	++	ND	++
Mφs	0	ND	+	ND	+
	0.3	ND	+	ND	+
	30	ND	++	ND	++
S iBALT	0	0	0	0	+
	0.3	0	0	0	++
	30	++	++	++	++

Eos—eosinophils; Neu—neutrophils; Mφ—macrophages; S iBALT—relative iBALT square on histological sections; + = positive effect, *p* < 0.05; ++ = positive effect, *p* < 0.01; 0 = no significant effect; ND = not determined in this study.

**Table 2 ijerph-19-13063-t002:** The effect of DEPs on the expression of transcripts of markers of antibody class switching and antibody production in the lung tissue and regional lymph nodes of immunized mice in comparison with mice immunized without DEPs with the indicated OVA doses.

Estimated Parameter	OVA Dose, µg	DEP1, 150 µg, L	DEP1, 150 µg, LN	DEP2, 150 µg, L	DEP2, 150 µg, LN
*germline ε*	0	+	0	0	++
	0.3	+	++	++	+
	30	+	0	+	++
*germline γ1*	0	++	+	++	+
	0.3	+	0	++	0
	30	+	0	+	+
*postswitch ε*	0	++	++	+	+
	0.3	++	+	++	−
	30	++	+	++	++
*postswitch γ1*	0	+	0	+	++
	0.3	+	+	0	++
	30	+	-	+	0
*circular µ-ε*	0	++	++	++	++
	0.3	++	++	++	++
	30	++	0	++	+
*circular γ1-ε*	0	0	+	++	++
	0.3	+	0	++	++
	30	0	0	0	0

L—lungs; LN—lymph nodes; + = positive effect, *p* < 0.05; ++ = positive effect, *p* < 0.01; - = negative effect, *p* < 0.05; 0 = no significant effect; ND = not determined in this study.

**Table 3 ijerph-19-13063-t003:** The effect of DEPs on CXCL13 production, proinflammatory cytokines, and the expression of B-cell-activation markers in the lung tissue and regional lymph nodes of immunized mice in comparison with those in mice immunized without DEPs with the indicated OVA doses.

Estimated Parameter	OVA Dose, µg	DEP1, 150 µg	DEP2, 150 µg
CXCL13	0	0	0
	0.3	0	++
	30	+	+
*Tnfa*	0	++	+
	0.3	+	++
	30	0	++
*Ifna1*	0	+	+
	0.3	+	+
	30	++	+
*Il4*	0	++	++
	0.3	++	++
	30	++	++
*Il13*	0	0	+
	0.3	++	+
	30	+	+
*Ifng*	0	−	0
	0.3	−	−
	30	0	−
*Cd19*	0	+	++
	0.3	+	+
	30	+	++
*Bcl6*	0	0	+
	0.3	0	+
	30	+	0
*Ebi2*	0	+	++
	0.3	+	+
	30	+	0

+ = positive effect, *p* < 0.05; ++ = positive effect, *p* < 0.01; − = negative effect, *p* < 0.05; 0 = no significant effect; ND = not determined in this study.

## Data Availability

The data presented in this study are available in figures within the article and in Appendix A.

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
