# Peer review of "DEPs Induce Local Ige Class Switching Independent of Their Ability to Stimulate iBALT de Novo Formation"

_ijerph, 2022, doi:10.3390/ijerph192013063_

Round 1

Reviewer 1 Report (Previous Reviewer 4)

The authors have addressed my concerns and therefore I would recommend the publication of the manuscript

Author Response

The authors have addressed my concerns and therefore I would recommend the publication of the manuscript.

Reply: We are very grateful to the reviewer for his previous comments and advices regarding our manuscript and for his recommendation for publication of our manuscript.

Reviewer 2 Report (Previous Reviewer 3)

The authors have addressed most of my comments from the original review, particularly tables summarizing their major findings and consistent use of blue and red colors for low and high OVA doses, respectively. The authors have also extensively edited their English language leading to better clarity and easier representation of their data. A few minor concerns have been highlighted in attached pdf copy of the resubmission. Please find below some general recommendations to enhance the quality of the work presented here: 

1) I recommend the authors to adhere to either antigen or OVA- and not keep interchanging it throughout the manuscript as this can be very confusing - given there are responses alone to Ag (OVA) and/or +/- DEPs. 

2) In the resubmission file, legends for some figures have been completely deleted, as an e.g., Fig.5, please rectify this.

3) Throughout the manuscript, some figures show data from DEPs used at 30 and 150ug while others focus only on 150ug DEPs. I strongly urge the authors to clearly indicate this accordingly in each respective figure legend as this is missing currently and is pertinent information.

Overall, the authors have addressed my concerns and discussed the potential impact of their study not just in the context of their findings but also present it as a potential model system to evaluate therapeutic paradigms for future studies in allergic airway inflammation. 

Author Response

The authors have addressed most of my comments from the original review, particularly tables summarizing their major findings and consistent use of blue and red colors for low and high OVA doses, respectively. The authors have also extensively edited their English language leading to better clarity and easier representation of their data. A few minor concerns have been highlighted in attached pdf copy of the resubmission. Please find below some general recommendations to enhance the quality of the work presented here:

Point 1. I recommend the authors to adhere to either antigen or OVA- and not keep interchanging it throughout the manuscript as this can be very confusing - given there are responses alone to Ag (OVA) and/or +/- DEPs.

Reply 1. We are grateful to the reviewer for his comments. In our new version of manuscript we changed “antigen” to “OVA” in most parts of the Abstract, Results and Discussion section where it was appropriate.

Point 2. In the resubmission file, legends for some figures have been completely deleted, as an e.g., Fig.5, please rectify this.

Reply 2. We are sorry for this misprint and corrected this in the new manuscript version.

Point 3. Throughout the manuscript, some figures show data from DEPs used at 30 and 150ug while others focus only on 150ug DEPs. I strongly urge the authors to clearly indicate this accordingly in each respective figure legend as this is missing currently and is pertinent information.

Reply 3. We have added the information about the usage of exactly high (150 µg) DEP doses in the legends of Figures 4-7 and S5-S6.

Overall, the authors have addressed my concerns and discussed the potential impact of their study not just in the context of their findings but also present it as a potential model system to evaluate therapeutic paradigms for future studies in allergic airway inflammation.

Reviewer 3 Report (Previous Reviewer 1)

Author has extensively revised the paper. It can be accepted in current format.

Author Response

Author has extensively revised the paper. It can be accepted in current format.

Reply: We are very grateful to the reviewer for his previous comments and advices regarding our manuscript and for his recommendation for publication of our manuscript.

Reviewer 4 Report (New Reviewer)

The authors are investigating the effect of Diesel exhaust particles (DEPs) on secondary lymphoid organs or locally (lungs and iBALT), in order to clarify the IgE class switching.  

The manuscript is clear and relevant for the field, being presented in a well-structured manner. The experimental design is appropriate to test the hypothesis. The manuscript results are reproducible based on the details provided in the methods section. The methods used are quite complex, from animal models, immunization and sample collection to histological analysis and H&E staining, ELISA, qPCR and flowcytometric evaluation of blood and tissue samples. To note that the marker introduced by the authors in the flowcytometric evaluation of blood cells, SiglecF, is very important for differentiating between nucleated cellular types.

All the animal experiments were performed according to an established and approved protocol, so the ethics regulations were fulfilled. 

The figures and schemes/images are appropriate and show the data properly, being easy to understand. The conclusions are consistent with the evidence and arguments presented. The cited references are relevant publications and do not include self-citations.

Author Response

The authors are investigating the effect of Diesel exhaust particles (DEPs) on secondary lymphoid organs or locally (lungs and iBALT), in order to clarify the IgE class switching. 

The manuscript is clear and relevant for the field, being presented in a well-structured manner. The experimental design is appropriate to test the hypothesis. The manuscript results are reproducible based on the details provided in the methods section. The methods used are quite complex, from animal models, immunization and sample collection to histological analysis and H&E staining, ELISA, qPCR and flowcytometric evaluation of blood and tissue samples. To note that the marker introduced by the authors in the flowcytometric evaluation of blood cells, SiglecF, is very important for differentiating between nucleated cellular types.

All the animal experiments were performed according to an established and approved protocol, so the ethics regulations were fulfilled.

The figures and schemes/images are appropriate and show the data properly, being easy to understand. The conclusions are consistent with the evidence and arguments presented. The cited references are relevant publications and do not include self-citations.

Reply: We are very grateful to the reviewer for his previous comments and advices regarding our manuscript and for such a high appreciation of our work.

This manuscript is a resubmission of an earlier submission. The following is a list of the peer review reports and author responses from that submission.

Round 1

Reviewer 1 Report

This research paper (DEPs induce local IgE class switching independent of their ability to 2 stimulate iBALT) is interesting and worthy of investigation. The overall paper reports some interesting results. But, I have some concerns regarding this manuscript.

11. The author should include more explanation regarding the practical benefits of DEPs in the introduction section.

22. What was the reason for using just female mice in the study. The random selection of animals is more preferable in such studies. Also, the author should put the details of housing and husbandry conditions.

33. The abbreviations of any term should be explained first, e.g., section 3.2 Immunization and Sample collection

44. On which base the dose rate was decided. Need to put references if the previous literature was followed or justify the dose.

55. The writing in study model is confusing, better to rewrite it.

66. Section 3.3. Gene expression measurements, Author should transfer primer sequencing into a table. It will be more appropriate.

77. Were any steps taken to minimize the effects of subjective bias when allocating mice to treatment (e.g., randomization procedure)? I didn’t notice such a description in the material and method section. Need to be added.

88. Describe the inclusion/exclusion criteria if animals were excluded from the analysis. Were the criteria pre-established?

99. Do the data meet the assumptions of the tests? (e.g., normal distribution)?

110. In the figure legends, define whether data describe technical or biological replicates.

111. Authors should identify potential research papers of methodology to refer.

112. The author should demonstrate the applicability of their procedure regarding major targets against atopic diseases in discussion section.

Author Response

Point 1. The author should include more explanation regarding the practical benefits of DEPs in the introduction section.

Response 1. We are very grateful to the reviewer for such an advice and we have added more references concerning the effect of DEPs on asthma development in clinical cases in the Introduction section.

Point 2. What was the reason for using just female mice in the study. The random selection of animals is more preferable in such studies. Also, the author should put the details of housing and husbandry conditions.

Response 2. We added the more detailed husbandry condition description in Secction 2.1. The reason for using female BALB/c mice was that such selection is standard for mostly currently used models of asthma on laboratory mice (). It is generally accepted that the intensity of humoral allergen-specific immune response and the symptoms of allergic inflammation in female mice is grater then in males (reference 1 and 2 in this comment for example).

  1. Takeda M., Tanabe M., Ito W., Ueki S., Konno Y., Chicara M., Itoga M., Kobayashi Y., Moritoki Y., Kayaba H., Chicara J. Gender difference in allergic airway remodelling and immunoglobulin production in mouse model of asthma. Respirol. 2013; 18(5); 797-806.
  2. Melgert B.N., Postma D.S., Kuipers I., Geerling M., Luinge M.A., van der Strate B.W.A., Kerstjensen H.A., Timens W., Hylkema M.N. Female mice are more susceptible to the development of allergic airway inflammation than male mice. Exp. Allergy. 2005; 35(11); 1496-15

Point 3. The abbreviations of any term should be explained first, e.g., section 3.2 Immunization and Sample collection.

Response 3. We added the explanation of the abbreviations in the section 3.2. and in the later sections.

Point 4. On which base the dose rate was decided. Need to put references if the previous literature was followed or justify the dose.

Response 4. We added the explanation and the references in the section 2.2. of the article.

Point 5. The writing in study model is confusing, better to rewrite it.

Response 5. We reordered the sentences and corrected English language in section 2.2. We also made a new figure (Figure 1 in the new version of article) to clarify the study design and immunization protocols used in the study.

Point 6. Section 3.3. Gene expression measurements, Author should transfer primer sequencing into a table. It will be more appropriate.

Response 6. We made a Supplementary Table 1 where the sequences of primers and probes is shown.

Point 7. Were any steps taken to minimize the effects of subjective bias when allocating mice to treatment (e.g., randomization procedure)? I didn’t notice such a description in the material and method section. Need to be added.

Response 7. Because in intact non-immunized mice of the same gender and age the main physiologic parameter associated with heath status is their weight we randomized mice in such way that in each group the average weight of animals did not differ significantly from other groups and amounted to 20-22 g. We added this in Secction 2.2.

Point 8. Describe the inclusion/exclusion criteria if animals were excluded from the analysis. Were the criteria pre-established?

Response 8. We added this information in Section 2.2 (First paragraph). Only mice without any signs of illness with weight more that 19 g were included in analyziz. if the animals showed signs of illness or significant depletion of the organism under the influence of the input components (exhaustion, poor condition of the coat, etc.), the animals were withdrawn from the experiment and analysis. All mice used in the experiments had no previous contacts with ovalbumin or DEPs.

Point 9. Do the data meet the assumptions of the tests? (e.g., normal distribution)?

Response 9. All data including in the final analysis meet the assumption of the ANOVA test. Only normal distributed data were included in ANOVA analysis. If some data were not normally distributed that was generally in the case of technical mistake until the mean values for each sample constituted a population that satisfies the criterion of normal distribution by Shapiro-Wilk test.

Point 10. In the figure legends, define whether data describe technical or biological replicates.

Response 10. The data in the figure legends indicate biological replicates. Each individual value was obtained by 2-3 technical replicates from individual mice and then generally mean and SD was estimated for the group of 6 mice.

Point 11. Authors should identify potential research papers of methodology to refer.

Response 11. In the methodology section in a new manuscript version authors refers to the necessary papers according to which the methodology was performed. DEPs doses were taken according to the research works:

Simon, D.A., Munoz, X., Gomez-Oloves, S., de Homdedeu M., Untoria, M.-D. Effects of diesel exhaust particle exposure on a murine model of asthma due to soybean. PLoS One. 2017; 12(6): e0179569.

De Grove, K.C., Provoost, S., Hendricks, R.W., McKenzie, A.N.J., Seys, L.J.M., Smitha Kumar, S., Tania Maes, T., Brusselle, G.G., 1, Joos, G.F. Dysregulation of type 2 innate lymphoid cells and TH2 cells impairs pollutant-induced allergic airway responses. J. Allergy Clin. Immunol. 2017; 139(1): 246-257.e4.

The antigen doses for intranasal immunization were taken according to our previous work with model based on subcutaneous immunization:

Chudakov, D.B., Rysantsev, D.Yu., Tsaregorodtseva, D.S., Kotsareva, O.D., Fattakhova, G.V., Svirshchevskaya, E.V. Tertiary lymphoid structure related B-cell IgEisotype switching and secondary lymphoid organ linked IgE production in mouse allergy model. BMC Immunol. 2020; 21: 45.

And on the basis of the fact that the amount of antigen that reached lungs after intranasal administration is about 2-3 times lower that administered one:

Southam, D.S., Dolovich, M., O’Byrne, P.M., Inman M.D. Distribution of intranasal instillations in mice: effects of volume, time, body position, and anesthesia. Am J Physiol Lung Cell Mol Physiol 2002; 282: L833–L839,

BAL fluid was analyzed according to reference:

Hoecke, L.V., Job, E.R., Saelens, X., Roose, K. Bronchoalveolar Lavage of Murine Lungs to Analyze Inflammatory Cell Infiltration. JoVe. 2017; 123: E55398.

ELSA and qPCR measurements were carried out by standard procedures described previously in our paper:

Chudakov, D.B., Rysantsev, D.Yu., Tsaregorodtseva, D.S., Kotsareva, O.D., Fattakhova, G.V., Svirshchevskaya, E.V. Tertiary lymphoid structure related B-cell IgEisotype switching and secondary lymphoid organ linked IgE production in mouse allergy model. BMC Immunol. 2020; 21: 45.

H&E histology was carried out with Abcam kit according to manufacturer instruction.

Point 12. The author should demonstrate the applicability of their procedure regarding major targets against atopic diseases in discussion section.

Response 12. In the model of DEPs induced allergic asthma established in this work we could observe the major components of allergic asthma presented in clinical cases, namely induction of specific and total IgE, infiltration of eosinophils, neutrophils and macrophages in BAL fluid, so our model is useful for evaluation of the effect of potential therapeutics on the key parameters of allergic asthma linked with type 2 immune response. We include the corresponding paragraph in discussion section. Also (the last paragraph in Discussion) we suggest that due to the ability of DEPs to induce IgE production without induction of germinal centers and iBALT formation such IgE production could be insensitive to the inhibitors of iBALT formation (for example inhibitors of TNFa) or germinal centers (Bcl6 inhibitors) also further works are required to clarify this suggestions.

Reviewer 2 Report

The manuscript by Chudakov's et al highlights the role of DEP in promoting the development of allergies, in particular by favoring the IgE class switch locally in the lung.

There are several observations and criticisms:

Although the experimental design is appropriate to test the hypothesis. The methods are not described in detail as an example the author claimed to have applied changes to some previously described method without clarifying the changes, so making the experiment not reproducible.

Title:

original: DEPs induce local IgE class switching independent of their ability to stimulate iBALT

I suggest a title more consistent with the study design and results:

DEPs induce local IgE class switching independent of their ability to stimulate iBALT de novo formation.

Major comments:

Methods:

State the number of animals and their allocation to the various analyses.

In line 195 the authors stated that “Each experiment was performed 2-3 times”.

Was each experiment done 2 or 3 times? possibly which experiments 2 times and which 3 times?

It is essential to clearly indicate the number of performed experiments

More important, does this mean that all the experiments (3.2, 3.3, 3.4, 3.5), for each animal, were repeated 2-3 times?

Was a mean value of the 3 experiments done for each animal? Then each mean value evaluated with the other means to formulate general mean and SD? or have all the values been considered in formulating the final means and SD?

In figures subheading, the authors stated that the picture represents 1 out of 3 experiments

Line 180: "slightly modified protocols used by other research groups"

It is essential to indicate what changes have been made.

Line 181-183: this phrase is a result and should be allocated in the specific section. In any case indicate whether macrophages, monocytes and dendritic cells are actually fewer than other cells (in this case report and discuss the result) or negligible number.

Results:

In general, all results are reported as discussion, without specific data, with bibliographic references of general statements.

Authors should report the data as detailed and clear as possible, so the interpretation of the data became easier to follow.

The study methods are reported also in some in figures.

 The result tables are reported in a supplementary excel file which is impossible to read and interpret. The essential values, at least the averages and SD, must be shown more clearly.

Lines 203-212: the sentences seem a repetition of what was reported in introduction and in discussion.

In the case the particles have been measured, it will be necessary to indicate the methods used for the measurements and the specific results in a dedicated subsection

Lines 212-213:

Indicate the values obtained do not state only comparable levels

Lines 214-216:

Language errors aside, why do you mention "after prolonged exposure"? does this mean that different results are obtained for shorter exposures?

From figure 1b (which we discuss below) it would seem that the increase in total IgE is minimal.

Fig 1:

the figure is overall confused and poorly interpretable, due to the superimposition of the many results (same consideration for figs 2, s1 and s2)

a)     sIgE titers means specific OVA IgE?

Legend line 226-228:  “one representative experiment out of three independent experiments” . If the author chose one experiment and excluded the others from a global evaluation, what rationale was used for this selection?

Similar observation in the other figs regarding the number of experiments

Lines 233-236

The author reported that IgA has an inhibitory power over IgE with a specific reference (27). But in this reference, and to my knowledge, no inhibitory activities of IgA on IgE production are described. The reference discuss the role of Treg cells in inhibiting the production of IgE and promoting IgA, with no reciprocal interaction between the two immunoglobulins.

Lines 238-239

The authors report an increase in IgG2 after immunization with OVA. This data should be discussed as IgG2a is typical of a Th1 response, as correctly indicated in line 234

Lines 264-266

The authors stated in line 181-183 that macrophage infiltration was minimal. From figure 2 it seems that macrophage number was higher than that of eosinophils.

Fig 2:

Red bars?

Lines 309-313:

Those are speculations.

The subsequent results have the same criticalities described. First of all, results are not detailed, but in part only discussed.

Lines 379-380: it is impossible to accept such statement without a proper discussion and interpretation.

Discussion:

Lines 464-469: these statements need explanation

All the discussion is confused and report speculations. Must be rewritten.

Minor comments:

Lavage fluid is BAL

There are many grammatical and spelling errors (lines: 75, 76, 99, 100, 480, 438-439, 567 etc…)

Line 184: 3.5

Lines 417-420: not clear

Line 462: longer

Lines 464-469: these statements need explanation

Lines 476-480: why results of the study justify the effect on DEP “mostly in people of middle or elderly ages or in people initially predisposed to atopic diseases”?

Line 495: Prohibited?

Line 496: This was than directly confirmed (what does it mean?)

Line 496-502: these statements should be rephrased.

Line 500: the accumulation of neutrophils was antigen-specific?

Line 568: “Common but not different properties for this two type of DEPs must be responsible for local B-cell IgE class switch induction”.

What does this mean? It seems obvious.

Author Response

Point 1. Although the experimental design is appropriate to test the hypothesis. The methods are not described in detail as an example the author claimed to have applied changes to some previously described method without clarifying the changes, so making the experiment not reproducible.

Response 1. We thanks to the reviewer for pointing out this shortcoming. In our new version of manuscript we added more detailed description of our key methods (ELISA, qPCR, histology) in the respective sections. We also pointed out that the main difference between our flow cytometry cell analyses of BAL fluid and the method described by L.V. Hoecke consisted in the application of additional marker for murine eosinophils identification (Siglec F). Our gating strategy is shown in Figure S3. Such description makes our modified method reproducible.

Point 2. Title: original: DEPs induce local IgE class switching independent of their ability to stimulate iBALT. I suggest a title more consistent with the study design and results: DEPs induce local IgE class switching independent of their ability to stimulate iBALT de novo formation.

Response 2, We thanks to the reviewer to this advice and changed the original title to a new one.

Point 3. Methods: State the number of animals and their allocation to the various analyses.

Response 3. We pointed out this in a new manuscript version. Initial groups contain 11-12 animals. After 8 weeks of immunization from these groups 5 mice were randomly selected for 2 weeks high OVA dose challenge and BAL fluid analysis. The remained mice were subjected to analysis of gene expression in lung tissue and regional lymph nodes by qPCR and analysis of antibody production by ELISA in serum samples. For H&E histology separate experimental series was performed with 6-7 mice in each group.

Point 4. In line 195 the authors stated that “Each experiment was performed 2-3 times”. Was each experiment done 2 or 3 times? possibly which experiments 2 times and which 3 times? It is essential to clearly indicate the number of performed experiments. More important, does this mean that all the experiments (3.2, 3.3, 3.4, 3.5), for each animal, were repeated 2-3 times? Was a mean value of the 3 experiments done for each animal? Then each mean value evaluated with the other means to formulate general mean and SD? or have all the values been considered in formulating the final means and SD? In figures subheading, the authors stated that the picture represents 1 out of 3 experiments.

Response 4. We performed Experiments with ELISA, qPCR and BAL cell analysis 3 times. Experiments with H&E histology were performed 2 times. For each separate experiment general mean and SD for all biological samples was calculated. All statistical significant differences obtained in each case and shown in the article was reproduced in these cases. However, because of the exact numerical results obtained in some analysis (antibody titers in ELISA, relative gene expression in qPCR) are usually slightly vary even for the same biological sample dependent on the exact reactive used (new volume of TMB substrate solution in ELISA, primary and secondary antibody samples in ELISA, new package of commercially manufactured probe in qPCR) and it is also impossible to made a set of all needed reactive for all series of all experiment (because of a long time needed for performance of separate experiment series and large quantities of reactive needed for all experiment) we decided to present in our manuscript the result from 1 of 3 (2) independent experiment. Although the exact numerical results (antibody titers in ELISA, relative gene expression in qPCR) were not exactly reproduced from one to another separate experiment the overall effects of DEPs and antigen doses on estimated parameters were reproduced (for example the combination of high DEP1 dose with low antigen dose always induced specific IgE production, germline ε transcripts and circular µ-ε transcripts expression in lungs tissue and lymph nodes, as well as circular γ1-ε transcripts expression in lungs but not in lymph nodes in comparison with mice immunized with low OVA dose alone; despite the fact that in one of the experiment the exact IgE antibody titers obtained for mice immunized with 0.3 µg OVA alone and for mice immunized with 0.3 µg OVA with high DEP1 dose were 22±7 and 650±250 respectively, and in the other case 30±4 and 780±320).

Point 5. Line 180: "slightly modified protocols used by other research groups" It is essential to indicate what changes have been made.

Response 5. In our new version of manuscript we pointed out that the main difference between our flow cytometry cell analyses of BAL fluid and the method described by L.V. Hoecke consisted in the application of additional marker for murine eosinophils identification (Siglec F).

Point 6. Line 181-183: this phrase is a result and should be allocated in the specific section. In any case indicate whether macrophages, monocytes and dendritic cells are actually fewer than other cells (in this case report and discuss the result) or negligible number.

Response 6. These cells subpopulations were actually present in BAL in negligible numbers. We pointed out this in the results section.

Point 7. In general, all results are reported as discussion, without specific data, with bibliographic references of general statements. Authors should report the data as detailed and clear as possible, so the interpretation of the data became easier to follow. The study methods are reported also in some in figures. The result tables are reported in a supplementary excel file which is impossible to read and interpret. The essential values, at least the averages and SD, must be shown more clearly.

Response 7. We added in the Results Sections 3.1, 3.2, 3.3. more detailed information about data obtained in this work and rewrote some paragraphs. The study methods reported in details in the Methods Section. We put a brief description of experiments in some figure signatures. We also improve our supplementary Excel File where we shown more clearly the data supporting the manuscript.

Point 8. Lines 203-212: the sentences seem a repetition of what was reported in introduction and in discussion. In the case the particles have been measured, it will be necessary to indicate the methods used for the measurements and the specific results in a dedicated subsection.

Response 8. We deleted the repeating fragment (line 205-212). In our study we have not measured particle size, the information about this in Introduction section is taken from NIST Certificates for these particles that have been added in the references list:  1.    Gonzales, C.A., Chocuette, S.J. National Institute of Standards and Technology. Certificate of Analysis. Standart reference material 1650B. Diesel particulate matter. Gaithersburg. Certificate Issue Date 07 July 2021. 2. Gonzales, C.A., Chocuette, S.J. National Institute of Standards and Technology. Certificate of Analysis. Standart reference material 2786. Fine Atmospheric Particulate Matter (Mean Particle Diameter <4 µm). Gaithersburg. Certificate Issue Date 02 July 2021.

Point 9. Lines 212-213: Indicate the values obtained do not state only comparable levels

Response 9. We complete this sentence.

Point 10. Lines 214-216: Language errors aside, why do you mention "after prolonged exposure"? does this mean that different results are obtained for shorter exposures? From figure 1b (which we discuss below) it would seem that the increase in total IgE is minimal.

Response 10. We clarify this sentence to: “High doses of DEPs in combined with low (Figure 1a, b) or high antigen doses (Figure S2a, b) in comparison to OVA alone induced formation of specific and total IgE in mice after prolonged administration8 weeks of immunization”. The increase of total IgE was not as marked as specific IgE in some groups but was significant.

Point 11. Fig 1: the figure is overall confused and poorly interpretable, due to the superimposition of the many results (same consideration for figs 2, s1 and s2); a) sIgE titers means specific OVA IgE? Legend line 226-228:  “one representative experiment out of three independent experiments” . If the author chose one experiment and excluded the others from a global evaluation, what rationale was used for this selection? Similar observation in the other figs regarding the number of experiments

Response 11. We clarify this figures to a new ones when the mean values in each group is marked by histogram charts. In this case the data marked by different colors does not overlap. We performed Experiments with ELISA, qPCR and BAL cell analysis 3 times. Experiments with H&E histology were performed 2 times. For each separate experiment general mean and SD for all biological samples was calculated. All statistical significant differences obtained in each case and shown in the article was reproduced in these cases. However, because of the exact numerical results obtained in some analysis (antibody titers in ELISA, relative gene expression in qPCR) are usually slightly vary even for the same biological sample dependent on the exact reactive used (new volume of TMB substrate solution in ELISA, primary and secondary antibody samples in ELISA, new package of commercially manufactured probe in qPCR) and it is also impossible to made a set of all needed reactive for all series of all experiment (because of a long time needed for performance of separate experiment series and large quantities of reactive needed for all experiment) we decided to present in our manuscript the result from 1 of 3 (2) independent experiment. Although the exact numerical results (antibody titers in ELISA, relative gene expression in qPCR) were not exactly reproduced from one to another separate experiment the overall effects of DEPs and antigen doses on estimated parameters were reproduced (for example the combination of high DEP1 dose with low antigen dose always induced specific IgE production, germline ε transcripts and circular µ-ε transcripts expression in lungs tissue and lymph nodes, as well as circular γ1-ε transcripts expression in lungs but not in lymph nodes in comparison with mice immunized with low OVA dose alone; despite the fact that in one of the experiment the exact IgE antibody titers obtained for mice immunized with 0.3 µg OVA alone and for mice immunized with 0.3 µg OVA with high DEP1 dose were 22±7 and 650±250 respectively, and in the other case 30±4 and 780±320).

Point 12. Lines 233-236 The author reported that IgA has an inhibitory power over IgE with a specific reference (27). But in this reference, and to my knowledge, no inhibitory activities of IgA on IgE production are described. The reference discuss the role of Treg cells in inhibiting the production of IgE and promoting IgA, with no reciprocal interaction between the two immunoglobulins.

Response 12. We rewrote this sentence to:” We also measured levels of specific IgG2a associated with type 1 immune response and IgA associated with Tregs activity in immunized mice”.

Point 13. Lines 238-239; The authors report an increase in IgG2 after immunization with OVA. This data should be discussed as IgG2a is typical of a Th1 response, as correctly indicated in line 234.

Response 13. We discussed this fact in new version of article in 10th paragraph of Discussion section in the connection with the absence of IFNγ expression induction in lung tissue by DEPs.

Point 14. Lines 264-266 The authors stated in line 181-183 that macrophage infiltration was minimal. From figure 2 it seems that macrophage number was higher than that of eosinophils.

Response 14. We clarify this sentence in a new version of manuscript to: “ Alveolar macrophages content in BALs wasware 2-3 times higher comparable in high and versus low dose groups immunized in combination with DEPs after short high dose challenge and was also enhanced by DEPs”

Point 15. Fig 2: Red bars?

Response 15. We are sorry for this mistake, red bars shows significant differences with mice immunized by high OVA dose alone; we corrected this sentence in figure legend.

Point 16. Lines 309-313: Those are speculations.

Response 16. We totally agree with the reviewer that this is speculative but this is the most probably explanation of why some Ig switch markers are induced by DEPs without any administered antigen.

Point 17. The subsequent results have the same criticalities described. First of all, results are not detailed, but in part only discussed.

Response 17. We have rewrote and complete some aragraphs in Results Section by more detailed description of results.

Point 18. Lines 379-380: it is impossible to accept such statement without a proper discussion and interpretation.

Response 18. From the overall results it is clear that two type of DEPs, despite some differences, had similar influence on pro-allergic antibody formation (IgE, IgG1), BAL cell composition and induction of local IgE class switching. We clarify this sentence to; “It is curious, why two closely related DEPs exerted different effects on iBALT formation while stimulating similar pro-allergic humoral response in terms of antibody production and induction of local Ig class switching.”

Point 19. Discussion: Lines 464-469: these statements need explanation

Response 19. In the groups of mice immunized by high antigen doses with DEP2 IgA titers although significantly higher in comparison with mice immunized only by high OVA doses without DEPs were about 2 orders in magnitude lower than IgG1 titers (corresponding to Type 2 immune response) in the same mice. We also pointed out later in Discussion section (in 10th paragraph where differences between DEP2 and DEP1 is discussed) that the induction of IgA in response to DEP2 may reflect only marginal Tregs induction due to the induction of some compensatory response at later stages that is dispensable and insignificant for inhibition of general inflammation. This is because of similar ability of two type of DEPs to induce pro-inflammatory cytokines expression (Figure 6 and Table 3 in a new version of manuscript).

Point 20. All the discussion is confused and report speculations. Must be rewritten.

Response 20. We exclude some paragraphs from Discussion that were speculative in some extent. We rewrote the other paragraphs and added some additional about BAL cell infiltration by different inflammatory cells as well as about differences and similarities of two type of DEPs with emphasize that it is similar but not different properties are linked with their ability to induce local IgE class switching an pro-allergic response (the later is shown emphatically in 11th  and 14th paragraphs of Discussion).

Point 21. Minor comments: Lavage fluid is BAL

Response 21. We substitute “lavage” to BAL in the article.

Point 22. There are many grammatical and spelling errors (lines: 75, 76, 99, 100, 480, 438-439, 567 etc…)

Response 22. We used the MDPI English editing service to correct our new version of manuscript.

Point 23. Line 184: 3.5

Response 23. We corrected this typo.

Point 24. Lines 417-420: not clear

Response 24. Sentence has been corrected in a new version of manuscript.

Point 25. Line 462: longer

Response 25. We corrected this.

Point 26. Lines 464-469: these statements need explanation.

Response 26. In the groups of mice immunized by high antigen doses with DEP2 IgA titers although significantly higher in comparison with mice immunized only by high OVA doses without DEPs were about 2 orders in magnitude lower than IgG1 titers (corresponding to Type 2 immune response) in the same mice. We also pointed out later in Discussion section (in 10th paragraph where differences between DEP2 and DEP1 is discussed) that the induction of IgA in response to DEP2 may reflect only marginal Tregs induction due to the induction of some compensatory response at later stages that is dispensable and insignificant for inhibition of general inflammation. This is because of similar ability of two type of DEPs to induce pro-inflammatory cytokines expression (Figure 6 and Table 3 in a new version of manuscript).

Point 27. Lines 476-480: why results of the study justify the effect on DEP “mostly in people of middle or elderly ages or in people initially predisposed to atopic diseases”?

Response 27. We deleted this speculative fact from the sentence.

Point 28. Line 495: Prohibited?

Response 28. The sentence was changed to: “This fact indirectly shows that the impact of DEPs on IgE response is not linked with germinal centers induction because these structures develops poorly when antigen is administered in low doses”

Point 29. Line 496: This was than directly confirmed (what does it mean?)

Response 29. We changed the sentence to; “This was than directly confirmed by the absence of significant germinal center marker Bcl6 induction in mice immunized with low doses alone or with low OVA doses with DEP1” to clarify the situation. It is also discussed in 11th paragraph of Discussion.

Point 30. Line 496-502: these statements should be rephrased.

Response 30. We re-wrote these statements. “So DEPs stimulate antigen-specific pro-allergic humoral response over a very wide range of antigen doses in contrast to situation observed in adjuvant-free model in our previous work”

Point 31. Line 500: the accumulation of neutrophils was antigen-specific?

Response 31. We deleted this sentence.

Point 32. Line 568: “Common but not different properties for this two type of DEPs must be responsible for local B-cell IgE class switch induction”. What does this mean? It seems obvious.

Response 32. This sentence needs to emphasis that common DEPs immunomodulating properties (the ability to induce type 2 cytokines expression) but not different ones (the ability to stimulate iBALT growing and germinal centers) are responsible for their effect on induction of local IgE class switching.

Reviewer 3 Report

In this study, the authors characterize local and systemic humoral immune responses to diesel exhaust particles (DEPs) either alone or in combination with low or high OVA immunization doses. The authors present evidence of direct and sequential class switching of IgE responses to DEPs and show the local humoral responses and class switching occurs in preexisting iBALT structures rather than those formed de novo. The manuscript is written in a logical manner with good data representation. The authors have also provided detailed general and statistical methods employed in this study. 

However, in several instances, the authors refer to the incorrect figure in their text. Furthermore, the colors used to represent low vs high OVA dose changes halfway through the manuscript, recommend to keep this consistent. I highly recommend to include a study design image to help readers understand their experimental paradigm and conditions tested. Furthermore, since the effects in terms of Ab titers, cellular composition in BAL, iBALT growing, etc. seem to be affected by a combination of both DEP concentration and most importantly OVA Ag dose, I strongly recommend to represent this in a table form for easier read and better understanding of the functional implications of their findings. 

The authors do not discuss much about how the cellular composition in BAL influences the cytokine responses detected in lung tissue and their influence of the observed results. For example, in lines 268-272, the eosinophil accumulation is very comparable between low and high dose OVA. Thus, I do not think this statement can be made. Instead the authors could speculate on the net effect of 7-8x higher macrophage accumulation in OVA low dose than neutrophil accumulation in OVA high dose. Also, difference between figure 5 and 4c is unclear, could there just be one set of representative images for the H&E staining?

Overall, the study is well carried out however, manuscript needs revision to better represent the major findings and fix incorrect figure references in the text. Also, in certain instances the authors have over interpreted their findings where existing data is insufficient to support such conclusions. 

Author Response

Point 1. In this study, the authors characterize local and systemic humoral immune responses to diesel exhaust particles (DEPs) either alone or in combination with low or high OVA immunization doses. The authors present evidence of direct and sequential class switching of IgE responses to DEPs and show the local humoral responses and class switching occurs in preexisting iBALT structures rather than those formed de novo. The manuscript is written in a logical manner with good data representation. The authors have also provided detailed general and statistical methods employed in this study. However, in several instances, the authors refer to the incorrect figure in their text. Furthermore, the colors used to represent low vs high OVA dose changes halfway through the manuscript, recommend to keep this consistent. I highly recommend to include a study design image to help readers understand their experimental paradigm and conditions tested. Furthermore, since the effects in terms of Ab titers, cellular composition in BAL, iBALT growing, etc. seem to be affected by a combination of both DEP concentration and most importantly OVA Ag dose, I strongly recommend to represent this in a table form for easier read and better understanding of the functional implications of their findings.

Response 1. We are grateful to the reviewer for high appreciation of our work and for these comments. We corrected the refers to the figures in the text. Also we re-maid some pictures so that in all of them groups of mice immunized without antigen are marked by gray color, groups immunized with low OVA dose are marked by blue color, and groups immunized by high antigen dose are marked by red one. We include a study design image in a new version (Figure 1). We also include 3 extra tables in our article where the effect of DEPs on different immunological parameters is shown.

Point 2. The authors do not discuss much about how the cellular composition in BAL influences the cytokine responses detected in lung tissue and their influence of the observed results. For example, in lines 268-272, the eosinophil accumulation is very comparable between low and high dose OVA. Thus, I do not think this statement can be made. Instead the authors could speculate on the net effect of 7-8x higher macrophage accumulation in OVA low dose than neutrophil accumulation in OVA high dose. Also, difference between figure 5 and 4c is unclear, could there just be one set of representative images for the H&E staining?

Response 2. We re-wrote this paragraph and added in Discussion section (5th paragraph in Discussion) more considerations about BAL cell content and about its linkage with type 2 immune response though it is not entirely clear what exact cell subpopulation was responsible for type 2 cytokines IL-4 and IL-13 as well as for TNFa production in our case. This question also very interesting was beyond the aim of this study. We also moved figure 5 to supplementary information (as Figure S6) to shorten the manuscript.

Point 3. Overall, the study is well carried out however, manuscript needs revision to better represent the major findings and fix incorrect figure references in the text. Also, in certain instances the authors have over interpreted their findings where existing data is insufficient to support such conclusions.

Response 3. We re-wrote some parts of results and discussion sections to more clearly representation of our results and the main findings from our work. We deleted some parts of Discussion section that seems to be too speculative. We also made a correction of our English style and grammar by means of MDPI English editing service.

Reviewer 4 Report

Chudakov et al. investigate in their study “DEPs induce local IgE class switching independent of their ability to stimulate iBALT” the connection between Diesel exhaust particles and IgE class switching. They administer small or medium sized Diesel particles in combination with OVA to mice and determine the production of immunoglobulins, cytokines and the formation of inducible bronchial-associated lymphoid tissue. Further they determine the localization of class switching by quantitative RT-PCR. The authors use suitable methods and include appropriate controls and the conclusions are supported by the presented data.

A few points should be clarified before publication:

·      y-axis labels in qPCR experiments: The authors use “deltaCt (norm)” which would indicate a substraction of Ct values or usage of delta delta Ct values, however they see differences up to 1000. It should be indicated, that most likely th 2delta delt Ct was plotted.

·      The paper contains some typos and should undergo another round of proof reading. (Such as line 114: 1.3 mL instead of 1.3 mL, line 48: brackets around citation 4 are missing).

·      Lines 198-201 contain instructions for the result section and should be deleted. 

Author Response

Point 1. Chudakov et al. investigate in their study “DEPs induce local IgE class switching independent of their ability to stimulate iBALT” the connection between Diesel exhaust particles and IgE class switching. They administer small or medium sized Diesel particles in combination with OVA to mice and determine the production of immunoglobulins, cytokines and the formation of inducible bronchial-associated lymphoid tissue. Further they determine the localization of class switching by quantitative RT-PCR. The authors use suitable methods and include appropriate controls and the conclusions are supported by the presented data.A few points should be clarified before publication: y-axis labels in qPCR experiments: The authors use “deltaCt (norm)” which would indicate a substraction of Ct values or usage of delta delta Ct values, however they see differences up to 1000. It should be indicated, that most likely th 2delta delt Ctwas plotted.

Response 1. We are grateful to the reviewer for high appreciation of our work and for these comments. We corrected the Y-axis labels if Figures 4, 6, 7 and S5. Indeed we plotted 2-Δ(ΔCt) values in the case of these experiments.

Point 2. The paper contains some typos and should undergo another round of proof reading. (Such as line 114: 1.3 mL instead of 1.3 mL, line 48: brackets around citation 4 are missing).

Response 2. We corrected these typos in the manuscript text.

Point 3. Lines 198-201 contain instructions for the result section and should be deleted. 

Response 3. We are sorry for this overlooking mistake. We have deleted this from the new version of manuscript.